

# Biogeochemical diversity and hot moments of GHG emissions from shallow alkaline lakes in the Pantanal of Nhecolândia, Brazil

Laurent Barbiero[1, 2, 3], Marcos Siqueira Neto[4], Rosangela Rodrigues Braz[1], Janaina Braga de Carmo[5], Ary Tavares Rezende Filho[6], Edmar Mazzi[1], Fernando Antonio Fernandes[7], Sandra Regina Damatto[8], Plinio Barbosa de Camargo[1]

[1]Laboratorio de Ecologia Isotópica, Centro de Energia Nuclear na Agricultura (CENA), Universidade de São Paulo, Piracicaba-SP, 13416-000, Brazil

[2]Instituto de Química, Universidade estadual de Campinas, Campinas-SP, 13083-970, Brazil

[3]Géoscience Environnement Toulouse (GET), IRD, CNRS, Université P. Sabatier de Toulouse, F31400, France

[4]Laboratorio de Biogeoquímica Ambiental, Centro de Energia Nuclear na Agricultura (CENA), Universidade de São Paulo, Piracicaba-SP, 13416-000, Brazil

[5]Centro de Ciências e Tecnologias para a Sustentabilidade, Universidade Federal de São Carlos, Sorocaba-SP, 18052-780, Brazil

[6]Faculdade de Engenharias, Arquitetura e Urbanismo e Geografia (FAENG), Universidade Federal do Mato-Grosso do Sul, Campo Grande-MS, 79070-900, Brazil

[7]Empresa Brasileira de Pesquisa Agropecuária (EMBRAPA), Embrapa-Pantanal, Corumbá-MS, 79320-900, Brazil

[8]Departamento de Metrologia das Radiações, Instituto de Pesquisas Energéticas e Nucleares (IPEN), Universidade de São Paulo, São Paulo-SP, 05508-000, Brazil

*Correspondence to*: Laurent Barbiero (laurent.barbiero@get.omp.eu)

**Abstract.** Nhecolândia is a vast sub-region of the Pantanal wetland in Brazil with great diversity in surface water chemistry evolving in a sodic alkaline pathway under the influence of evaporation. In this region, more than 15,000 shallow lakes are likely to contribute an enormous quantity of greenhouse gas to the atmosphere, but the diversity of the biogeochemical scenarios and their variability in time and space is a major challenge to estimate the regional contribution. In this study, we compiled measurements of the physico-chemical characteristics of water and sediments, gas fluxes in floating chambers, and sedimentation rates to illustrate this diversity. Although these lakes have a similar chemical composition, the results confirm an opposition between the black-water and green-water alkaline lakes, corresponding to distinct biogeochemical functioning. Black-water lakes are $CO_2$ and $CH_4$ sources, with fairly constant emissions throughout the seasons. Annual carbon dioxide and methane emissions approach 790 mmol m$^{-2}$ y$^{-1}$ and 73 mmol m$^{-2}$ y$^{-1}$, respectively. By contrast, green-water lakes are $CO_2$ sinks but significant $CH_4$ sources with fluxes varying significantly throughout the seasons, depending on the development of the cyanobacterial bloom. The results highlight two hot moments for methane emissions. The first one is suspected after the disappearance of the cyanobacterial bloom, which is accompanied by a drop in pH of the upper part of the sediments. The second one is identified when the $O_2$-supersaturation is reached under extreme bloom and sunny weather conditions, which provoke an abrupt $O_2$ purging of the lakes. Taking into account the seasonal variability, annual methane emissions are around 8,850 mmol m$^{-2}$ y$^{-1}$, i.e., much higher than reported in previous studies for alkaline lakes in





Nhecolândia. Carbon dioxide consumption is estimated about 1,140 mmol m$^{-2}$ y$^{-1}$. However, these balances must be better constrained with systematic and targeted measurements around these hot moments.

## 1 Introduction

Wetlands contribute to the creation of large reservoirs of biodiversity, improve the quality of surface water, reduce flood risk
associated with extreme rainfall, and supply streams during low water periods (Brinson et al., 1981; Fustec and Lefeuvre, 2000; Mitsch and Gosselink, 2015; Reddy and DeLaune, 2008; Turner, 1991; Whiting and Chanton, 2001). They are also the site of production and emission of greenhouse gases (GHG), which depend on many factors, including pH, oxidation-reduction conditions, and biogeochemical functioning (Bastviken et al., 2004, 2011; Wang et al., 1993). An important source of uncertainty on the GHG budget is attributable to emissions from wetland and other inland waters (Saunois et al., 2016). In
this context, the development of process-based models for inland water emissions, constrained by observations, is still regarded as a priority. Among the wetlands, tropical wetlands are known to be highly reactive, as permanent high temperatures increase the velocity of the biogeochemical reactions (Fustec and Lefeuvre, 2000; Reddy and DeLaune, 2008). As an additional restriction, continental alkaline wetlands are characterized by an increase in water pH during evaporation, favoring the solubilization, transfer and accumulation of organic matter in the landscape. Collectively, these conditions can
lead to highly reactive portions of landscape; i.e. emission variability in time and space where the greenhouse gas fluxes are poorly constrained (Bogard et al., 2014; Peixoto et al., 2015).

Nhecolândia is a sub-region of the Pantanal wetland in Brazil, where a myriad of shallow saline-alkaline, oligosaline and freshwater lakes and ponds coexist in the landscape, sometimes at short distances from each other (~200 m). The number of lakes was estimated from 12,000 to 17,500 including 500 to 600 saline-alkaline ones (Costa et al., 2015; Evans and Costa,
2013). Furian et al. (2013) have shown that the level of salinity of neighboring lakes are independent and that the co-existence of both, saline and freshwater lakes occurs as a result of the distribution of the soil cover that induce a differential hydrological regime. Under the influence of cumulative evaporation over the years, the pH of some saline lakes has reached high values, close to or above 10, resulting in an increasing solubility of the organic matter, with dissolved organic carbon (DOC) values up to 750 mg L$^{-1}$ (Barbiero et al., 2016; Mariot et al., 2007). Martins (2012) noticed that waters of neighboring
saline lakes with almost similar chemical composition can have permanent black or green color, resulting from opposite biogeochemical functioning, but the parameters that control such differences are still poorly understood. Collectively, the size of the region, the number of lakes and diversity of biogeochemical conditions in space and time (day-night and seasonal change) make it difficult to estimate the regional greenhouse gas emissions from Nhecolândia. A prerequisite for such a regional balance and its contribution to the global budget is a better understanding of the diversity of scenarios and of their
variability in time and space (Peixoto et al., 2015). The aim of this study is precisely to present this diversity in the specific context of Nhecolândia and to provide preliminary results of the range of greenhouse gas fluxes from the most alkaline surface water.



## 2 Materials and Methods

### 2.1. Studied area

The Pantanal wetland, considered the largest wetland of the world (Por, 1995), is an immense tropical floodplain located between 15° and 22°S and 55° and 60°W (Fig. 1). This depression is supplied and drained by the Paraguay River and its

tributaries, most of them joining on the eastern bank, from the surrounding Brazilian mountains and plateaus. Characterized by low altitudes (100-200 m) and extremely low topographical gradients (0.02 to 0.03°), the alluvial plain is partially covered by seasonal flooding mainly in summer (from November to March) exerting a marked modulating effect on the Paraguay River hydrology (Junk and Nunes de Cunha, 2005). At the regional scale, this wetland consists of a mosaic of sub-wetlands that usually differ in various aspects, i.e. hydrological, geomorphological, pedological and geochemical (Rezende-

Filho et al., 2012; 2015). Nhecolândia is one of the largest (24,000 km$^2$) sub-regions delimited by the Taquari River in the north and northwest, a portion of the Paraguay River in the west, the Negro River in the south, and by a highland plateau in the east, called Serra de Maracajú (Fig. 1). The region is characterized by an abundance of small, shallow (0.5 to 1.5m deep), non-stratified, round or irregular-shaped lakes and ponds (Pott and Pott, 2011). The co-existence of freshwater and saline lakes occurs mainly in the southern and south western, lowland portions of Nhecolândia. Furian et al. (2013) have shown

that the presence of salinity in the lakes results from an ongoing process of accumulation and evaporation under relatively humid climate but poor drainage conditions. These authors have determined that the evaporation process and associated mineral precipitations that control dissolved Ca, Mg and K account for at least 86% of the chemical variability in surface waters (n = 147 lakes) based on major ion chemistry. According to Rezende Filho et al. (2012), the gradual and continuous chemical changes from the most diluted to the most concentrated waters allowed to link their origin to the chemistry of the

Taquari River freshwater that supply the region. Nhecolândia has relatively closed drainage with little connection to major fluvial systems, and therefore, the organic matter in the lakes depends more on the local cycles than on terrestrial inputs by the annual inundation pulse (Mariot et al., 2007). Semihumid climate patterns, classified as tropical humid with short dry season ("Aw" type in the Köppen classification), are controlled by the seasonal migration of the Intertropical Convergence Zone (ITCZ). The mean annual air temperature is ~25 °C ranging from 21 °C during dry winters to 32°C during rainy

summers. The Mean annual precipitation (P) is ~1100 mm, and the annual evapotranspiration (ETP) is approximately 1400 mm, providing a hydrological deficit of about 300 mm. At the local level, the strong thermal contrast between saline lakes and surrounding forested area causes low day–night alternating winds that enhance evaporation (Quénol et al., 2006). Previous studies have confirmed a high $CH_4$ flux from the Pantanal wetland, but have focused on spatial variability within lakes or have been limited through the use of short measurement periods throughout the year (Bastviken et al., 2010; Marani

and Alvalá, 2007). Other recent studies have been performed out of Nhecolândia, i.e. out of the specific chemical context with a wide range of alkaline salinity (Peixoto et al., 2015). In Nhecolândia, the water shows a positive calcite residual alkalinity and evolves in a sodic-alkaline pathway under the influence of the evaporation (Barbiero et al., 2002; 2004), leading to high pH values. In some saline lakes, the high pH conditions are adverse to many phytoplankton organisms, but



may support growth of dense alkaliphylic cyanobacterial blooms. Usually, the most extreme cyanobacterial blooms are found during the driest period (August to November), with the highest alkalinity, EC and temperature values (Santos et al., 2011). Two major bloom-forming cyanobacterium have been identified (Andreote et al., 2014; Costa et al., 2016; Genuário et al., 2017; Vaz et al., 2015), *Anabaenopsis elenikinii* and *Arthrospira platensis*, primary producer (halo)alkaliphilic oxygenic cyanobacteria (Sorokin et al., 2015) that fix inorganic carbon and produce oxygen. These blooms occur during the dry season and disappear in a few hours after the first heavy rains of the wet season.

The study was carried out in two different regions of Nhecolândia, in 4 lakes at the Centenário farm (private land), and 2 lakes at the Nhumirím farm that belong to the Brazilian Agricultural Research Corporation (Fig. 1). Data were acquired on one freshwater lake (F) and five saline-alkaline lakes (M, V, P, G and R) at different periods of the year from 2012 to 2015. Because of flooding that makes it impossible to access the site, no data were collected in the height of the wet season, and fieldwork was concentrated in the early (May-June), medium (July-September), and in the late dry season (October-December). The lakes were selected so as to cover a wide range of water electrical conductivity, and according to the presence and intensity of cyanobacterial blooms. The gas emission data were acquired from lakes M, V, P and G during 24-hour cycle monitoring (usually every 2 hours) and are supplemented by data acquired occasionally but systematically in each field campaign.

## 2.2. Study Design and Analytical Methods

### 2.2.1. Gas emission study design

Gas fluxes from the lake to the atmosphere were measured using 32-L polyethylene floating chambers, having a base area of $0.195 \ m^2$. The main conditions during the field campaigns are summarized in table 1. Two procedures were used for these measurements with static or dynamic chambers. For the static procedure, each chamber was anchored with a 10-m line to avoid drifting, and located from the center to the border of the lake. For the dynamic one and depending on the lake diameter, a train of 3 to 6 floating chambers was attached, leaving a gap of 10 meters between two successive floating chambers. Floating chambers were placed in the water every minute from the lake shore, and then slowly pulled toward the opposite bank at a rate of about 5 m min$^{-1}$. This experimental design allows for scanning the various water column heights, with the least turbulence disruption to the lake surface. To minimize artificial turbulence effects, foam elements were adjusted so that a maximum of 2 cm of the chamber penetrated below the water surface. Due to the low water column, it was not possible to place a bubble shield to prevent bubbles from reaching the chamber. Therefore, the results represent the sum of both fluxes by diffusion and ebullition. Gas samples were collected through a 60-mL syringe after 20 min by static procedure, and once each chamber reached the opposite bank for the dynamic procedure i.e. after 20 to 35 min, depending on the lake diameter. Samples were collected in 2 replicates for each chamber. The gas samples were transferred into 30-mL glass bottles, previously capped with gas-tight, 10-mm thick butyl rubber septa and aluminum caps, and evacuated with a hand vacuum pump. Air samples were also collected at the departure of the chamber train for the ambient gas levels. Gas





concentrations in the liquid phase were estimated indirectly using a headspace displacement method (Hope et al., 1995) with a 120-mL syringe and an air:water volume proportion of 1:3 (30:90 mL). To equilibrate the headspace with the liquid phase, the syringe was shaken for 2 min by hand before injecting the headspace gas into the 30-mL glass bottle. Gas concentrations ($CH_4$, $CO_2$ and $N_2O$) were measured by gas chromatography model Shimadzu GC-2014 (Shimadzu Co., Columbia, MD,

USA). The chromatographer was equipped with a packed column, an electron capture detector (ECD) to analyze $N_2O$, and a flame ionization detector (FID) to quantify $CO_2$ and $CH_4$. Prior to detection, $CO_2$ was reduced to $CH_4$ using a methanizer. The gas analyzer was calibrated with certified $CO_2$ (270 and 2090 ppm), $CH_4$ (690 and 3150 ppm) and $N_2O$ (0.315 and 1.200 ppm) gas standards (minimum and maximum, respectively). The analyses were performed in the Environmental Science Laboratory (UFSCar, Sorocaba, Brazil). Gas fluxes were calculated by the linear change in the amount of gas in the

chambers as a function of sampled time.

### 2.2.2. Biogeochemical field indicators

A set of basic physico-chemical data was collected in water (pH, oxidation-reduction potential (mV), electrical conductivity ($\mu S\ cm^{-1}$), temperature (°C), dissolved $O_2$ saturation (%), turbidity (NTU)), and sediments (pH and oxidation-reduction potential) as well as air temperature inside and outside the floating chambers. To obtain the oxidation-reduction potential, a

value of +203 mV was added to the Pt-probe measured potential, assuming that the temperature was almost constant close to 30 °C. Chlorophyll-a and phycocyanin were measured with a Trilux fluorometer (Chelsea Technologies Group) with excitation wavelength at 470 and 610 nm, respectively, in order to estimate the cyanobacteria biomass.

### 2.2.3. Water chemistry

Water samples were collected from lakes F, M, V, G, R and P and centrifuged in the field (15 min at 6000 rpm: Relative

Centrifuge Force = 3200 g) in order to remove algae and other suspended particles. The supernatant was filtered with a 0.45μm cellulose acetate syringe filter. Filtered samples were subsequently stored in previously acid washed HDPE 125 ml containers and stored at 4°C in dark conditions until analysis. Major anions ($F^-$, $Cl^-$, $SO_4^{2-}$) and cations ($K^+$, $Na^+$, $Ca^{2+}$, $Mg^{2+}$) were measured by ion chromatography, alkalinity by 0.1N HCl titration, and dissolved organic carbon (DOC) concentrations were determined with a Shimadzu TOC-5000A. In order to compare the collected soil solutions with the regional chemistry

presented by Furian et al. (2013), the results of the water analysis were plotted into concentration diagrams based on their sodium amount.

### 2.2.4. Sedimentation rate

Vertical sediment cores were collected from lakes V, G and P using a one-inch polyethylene tube in order to determine the age of the sediments and to identify any drastic change in the rate of sediment deposition in the lakes over the last 100 years.

The age and the sedimentation rate determination were achieved by the unsupported [210]Pb method (half-life 22.3 years).



Each core was sliced into 2-cm sub samples, sifted in 0.09-mm sieves (170 mesh) with Milli-Q water, oven dried at 60°C and homogenized in a glass mortar. A 1-g aliquot of each slice was dissolved in acid ($HNO_3$ (65%), HF (40%)) and $H_2O_2$ 30% in a microwave digester and submitted to a sequential procedure for radiochemical determination of $^{226}Ra$ and $^{210}Pb$. The procedure consists of an initial precipitation of Ra and Pb with $3M\text{-}H_2SO_4$, dissolution in nitrile tri-acetic acid at basic

pH, precipitation of Ba ($^{226}Ra$)$SO_4$ with ammonium sulfate, and $^{210}PbCrO_4$ by sodium chromate. The $^{226}Ra$ concentration was assessed measuring the alpha counting on the precipitate Ba($^{226}Ra$)$SO_4$, whereas the $^{210}Pb$ concentration was assessed through its disintegration product $^{210}Bi$, measured by beta counting on the precipitate $^{210}PbCrO_4$ (Moreira et al., 2003). Unsupported $^{210}Pb$ (denoted $^{210}Pb_{Exc}$) were obtained from the $^{226}Ra$ and total $^{210}Pb$ activities. Radionuclides were measured in a low background gas-flow proportional counter at IPEN (São Paulo). Ages and sedimentation rates were calculated using

the CRS (Constant Rate of Supply) model (Appleby and Oldfield, 1978).

## 3 Results

### 3.1. Water major ion chemistry

The water chemical composition of the studied lakes is plotted in Fig. 2, where the solid lines denote the regional trends of

the major ions versus the sodium concentration drawn from the analyses of 147 lakes (Furian et al., 2013). Our results cover a wide range of salinity with $Na^+$ concentration from 0.3 to 500 mg $L^{-1}$. Only one freshwater lake was sampled in this study. Therefore, it was not possible to assess the major ions trends for low concentrations below $Na^+$ values of about 20 $mmol_c$. For higher concentrations, alkalinity and $K^+$ increased less rapidly than sodium amount. The $Ca^{2+}$ and $Mg^{2+}$ values decreased whereas the trends of $Cl^-$ and $SO_4^{2-}$ were not so clear with widely dispersed scatter plots, but their amounts were roughly in

agreement with the regional trends. The ionic proportions were substantially similar for lakes with green or black water. It should be noted, however, that during these 3 years of monitoring, the salinity range observed in green-water lakes (700 $\mu S$ $cm^{-1}$ < EC < 35,000 $\mu S$ $cm^{-1}$ and 2.5 $mmol_c$ < $Na^+$ < 500 $mmol_c$) was larger than that observed in black-water lakes (500 $\mu S$ $cm^{-1}$ < EC < 9,800 $\mu S$ $cm^{-1}$ and 2 $mmol_c$ < $Na^+$ < 102 $mmol_c$), but the two EC ranges overlapped significantly over a factor of 12 (700 $\mu S$ $cm^{-1}$ < EC < 9,800 $\mu S$ $cm^{-1}$).

### 3.2. Indicators of biogeochemical functioning

Field parameters for 24-h monitoring are presented in Fig. 3. The results clearly highlight two opposite biogeochemical functions in saline alkaline lakes, associated with black or green waters. Such opposition is permanent, since the black water lakes never presented cyanobacterial blooms, whereas green water lakes showed regular blooms throughout the dry seasons that disappeared quite abruptly (within a few hours) after each significant rain. For black water lakes, the changes in the

biogeochemical indicators over 24 hours were moderate. Electrical conductivity, pH and turbidity were stable, showing only slight variations. The turbidity values were within the range of 100 to 350 NTU depending on the season. The daily temperature oscillations were about 5 to 6°C amplitude. Dissolved oxygen saturation evolved in a range of 80 % to 120 %



during night and day time, respectively. Similarly, changes in EC, pH and turbidity in green water lakes were fairly moderate between day and night. However, deep differences were observed depending on the stage of development of the cyanobacterial bloom, which made it possible to distinguish 3 cases: (i) in the absence of bloom, the water turbidity remained moderate (<20 NTU), the cyanobacteria biomass was below 100 µg L$^{-1}$, the temperature fluctuated in a range of about 10 degrees (24-34 °C) with a minimum at about 6 a.m., and a maximum at 2 p.m. Dissolved oxygen saturation varied from a minimal value of about 3 % at the end of the night, and reached about 120 % at 2 p.m.; (ii) with a moderate bloom, the turbidity and the daily temperature oscillation were usually in the range of 500-800 NTU and 23-36 °C, respectively. The cyanobacteria biomass was over-range (> 120 µg L$^{-1}$) for our field equipment. The dissolved $O_2$ saturation values dropped below 1 % during the night, and increased to 300 or 350% (20 to 25 ppm) from 2 to 4 p.m. Then, the values decreased regularly to fall below 100 % around 9 p.m., then below 1 % around midnight; (iii) when the bloom was strong, the turbidity values exceeded 1000 NTU (up to 3500 NTU), the cyanobacteria biomass was over-range and the daily temperature fluctuations were about 20 °C (23 to 42 °C). During the day and in the absence of clouds, dissolved $O_2$ saturation values increased very rapidly, exceeding 500 % around noon. These values then exceeded the technical range of the equipment, but they continued to increase progressively and values up to 800% (~ 55 ppm) have been noted at about 1:30 p.m. Then, within a few minutes the values dropped down to 300% (~ 20 ppm), accompanied by a generalized ebullition of the lake. The bubbles were generally less than 0.1 mm in diameter and formed within the whole water column. When the bloom was strong but the weather cloudy, fluctuations similar to a case with moderate bloom were observed.

### 3.3. Sediment pH and oxidation-reduction conditions

The pH and oxidation-reduction conditions in the lake sediments varied respectively from 5.7 to 6.9 and -50 mV to +460 mV in the freshwater lake, 7.4 to 8.9 and -200 mV to +250 mV in the black water saline lakes and 9.2 to 10.2 and -360 mv to -50 mv in the green water saline lakes. In September 2014, a 150-mm rainstorm within 3 hours caused a significant dilution in green water lake G. The water electrical conductivity dropped from 35,000 µS cm$^{-1}$ to 3,600 µS cm$^{-1}$, while the pH ranged from 10.4 to 10.3. We noted that the bloom disappeared and massive amounts of algal organic matter were deposited, causing low oxidation-reduction potentials at -400 mV, whereas the pH dropped from 9.8 to 7.8 in the upper part of the sediment (arrow on Fig. 4). These conditions were maintained for several days.

### 3.4. Sedimentation rates

The mean sedimentation rates for the cores obtained through $^{210}$Pb dating method were 0.41 cm Y$^{-1}$ for the black water lake P, and 0.77 cm Y$^{-1}$ and 0.24 cm Y$^{-1}$ for green water lakes G and V respectively. Although the sedimentation rate for lake P was rather constant, that of lake G was much more scattered around the mean value (Fig. 5). For Lake V, the results seem to indicate the presence of two periods with a sedimentation rate of 0.1 cm y$^{-1}$ for the period 1945-2010 and 0.56 cm y$^{-1}$ for the period 1920-1945.



### 3.5. Gas emission

Gas emissions vary significantly throughout 24-hour cycles and depending on the type of lake. Figure 6 shows the $CH_4$ concentration in the lake water calculated from the headspace measurements. For green water lakes, the mean values were of 0.64 µmol $L^{-1}$ without bloom, and 4 µmol $L^{-1}$ and 56 µmol $L^{-1}$ for a moderate and strong bloom, respectively. All the lakes

showed a supersaturation with respect to the atmosphere. For black water lakes, $CH_4$ concentrations were much lower with a mean value of 0.12 µmol $L^{-1}$. Figure 7 shows the $CH_4$ emissions obtained on the green water lakes M, G, and V. In the absence of bloom (2 campaigns), an average $CH_4$ emission of 0.7 mmol $m^{-2}$ $d^{-1}$ was calculated, being slightly higher during the night (1.2 mmol $m^{-2}$ $d^{-1}$). The differences in the emission values between the floating chambers were moderate. On the other hand, $CO_2$ measurements (Fig. 8) indicate a consumption of 1.6 mmol $m^{-2}$ $d^{-1}$, higher during the day (3 to 4 mmol $m^{-2}$

$d^{-1}$) than during the night (0.5 to 1 mmol $m^{-2}$ $d^{-1}$). At the beginning of the cyanobacterial bloom (1 campaign), similar trends were observed with higher $CH_4$ emissions of about 6 mmol $m^{-2}$ $d^{-1}$ overnight and 2 mmol $m^{-2}$ $d^{-1}$ at the peak of the day (Fig. 7). Because of moderate $CH_4$ bubbling from the sediment, the differences between the chambers are higher depending on the amount of bubbles reaching the chambers during their movement across the lake. The consumption of $CO_2$ increased significantly up to 15 to 20 mmol $m^{-2}$ $d^{-1}$ during the day and 2 to 5 mmol $m^{-2}$ $d^{-1}$ at night. With a stronger cyanobacterial

bloom established for several weeks (1 campaign), methane emissions increased, reaching values close to 75 mmol $m^{-2}$ $d^{-1}$. Finally, during the afternoon at the beginning of the lake ebullition, $CH_4$ emissions shifted abruptly up to about 600mmol $m^{-2}$ $d^{-1}$ and such values were maintained at least until 6 p.m. Meanwhile, the consumption of $CO_2$ was about 25 mmol $m^{-2}$ $d^{-1}$, decreasing to 8 mmol $m^{-2}$ $d^{-1}$ at the beginning of the night. Unfortunately, because of a storm, the monitoring was interrupted as it was not possible to continue the measurements overnight.

The methane fluxes measured from the black water lake P (3 campaigns) were significantly lower with a mean value of 0.18 mmol $m^{-2}$ $d^{-1}$ and a standard deviation of 0.05 mmol $m^{-2}$ $d^{-1}$ (Fig. 9). The low standard deviation values between the chambers for each series of measurement highlighted the near absence of $CH_4$ bubbling from the sediments. The emissions of $CO_2$ were higher, from 1mmol $m^{-2}$ $d^{-1}$ to 5 mmol $m^{-2}$ $d^{-1}$ during the night and day, respectively, and a mean value of 2.16 mmol $m^{-2}$ $d^{-1}$. Large variations were observed between the chambers. $N_2O$ measurements indicated consumption from 0 to 2

µmol $m^{-2}$ $d^{-1}$ for the no-bloom and moderate-bloom conditions, with a trend to increase up to 7 µmol $m^{-2}$ $d^{-1}$ when the bloom was strong (Fig. 10). For the black water lake, no clear trend of $N_2O$ fluxes was observed (not shown).

## 4 Discussion

### 4.1. Diversity of surface water

The water chemistry (Fig 2) of the studied lakes covers a large range of alkaline salinity as observed at the regional level by

Furian et al. (2013). Therefore, from the point of view of mineral chemistry, the lakes are representative of the region and evolve in a sodic-alkaline pathway under the influence of evaporation (Barbiero et al., 2004). In this context, the





geochemical processes occurring in the lakes are known. With increasing salinity, $K^+$ is withdrawn from the solution to form Fe-mica (Furquim et al., 2010a; Barbiero et al., 2016), whereas alkalinity is involved in the formation of other clay minerals and calcite. The cations $Ca^{2+}$ and $Mg^{2+}$ are controlled by the formation of Mg-calcite with about 5 % Mg (Barbiero et al., 2008) and Mg-smectites of stevensite and saponite types (Furquim et al., 2008; 2010b). Since the waters evolve in a sodic-

alkaline pathway, the salinity range is associated with a wide range in pH values, which affects both surface water (5.5 < pH < 10.5) and lake sediments (5.7 < pH < 9.9). However, the range of alkaline salinity is not the only axis driving the diversity of this wetland. Consistent with Martins (2012), the results confirmed that the functioning of black and green water lakes differs significantly, despite a very close mineral chemistry throughout the season (Fig. 3). Although we can expect a higher accretion rate for the sediments that are seasonally supplied by the organic matter from the cyanobacteria blooms, the

sedimentation rates do not seem to reflect this difference in biogeochemical functioning (Fig. 5). The rates are relatively similar for black or green water lakes and also very similar to those presented by Fávaro et al. (2006) from a core collected on Lake M (0.61 cm $Y^{-1}$) using the same methodology. The low sedimentation rates observed for the recent sediments in Lake V are difficult to interpret. By contrast, the oscillations observed for Lake G may be due to its shallow depth that causes this lake to dry out during the most severe droughts and cause large cracks to appear in the sediment. The fall of more

recent material from the top down to the depth between the prisms may explain the results obtained for lake G.

The difference in biogeochemical functioning has implications for GHG emissions. During our campaigns, black water lake P was low $CO_2$ and $CH_4$ sources (2.16 mmol $m^{-2}$ $d^{-1}$ and 0.18 mmol $m^{-2}$ $d^{-1}$, respectively), and these values seem to be almost constant throughout  the year. It seems that blackwater lakes are much less reactive than those with green water, which were clearly $CH_4$ sources and $CO_2$ and $N_2O$ sinks. However for the green water lakes, methane emissions varied largely

depending on a combination of factors, i.e. the intensity of the cyanobacteria bloom, sunshine and significant rainfall.

### 4.2. Specificities of green water alkaline lakes

In the absence of bloom, dissolved methane concentrations were of the order of 0.7 μM, i.e. slightly higher than reported by Bastviken et al. (2010) for lake M during a field campaign in 2009 (0.17 μM) and in moderate bloom conditions (lake M is referred to by these authors as lake N7a). The $CH_4$ fluxes measured by these authors were of about 1.42 mmol $m^{-2}$ $d^{-1}$, with

54 and 46 % of ebullitive and diffusive fluxes, respectively. We obtained roughly similar fluxes with an average value of 0.78 mmol $m^{-2}$ $d^{-1}$. However, our data show that the water methane concentration increases with the intensity of the bloom. This trend has already been observed in the oxic layers of oligotrophic lakes, where positive correlations between oxic−water−methane and chlorophyll concentrations suggest an oxic methane production associated with primary production (Grossart et al. 2011; Bogard et al. 2014; Tang et al. 2014; 2016). According to Tang et al. (2016), microorganisms may

produce methane by being equipped with enzymes to counteract the effects of molecular oxygen during methanogenesis or using alternative pathways that do not involve oxygen-sensitive enzymes. In this context and with a moderate bloom, the water $CH_4$ concentrations increased up to 4.26 μM but these values remained within the typical range of 0.01 to 10 μM observed in many surface freshwaters including previous measurements in the Pantanal (Bastviken et al., 2010; Marani and





Alvalá, 2007). This is no longer the case, however, in the presence of a strong bloom with dissolved $CH_4$ contents reaching about 60 μM. Methane fluxes increase considerably up to 75 mmol m$^{-2}$ d$^{-1}$ (a single measurement with 3 floating chambers) in the morning. During the afternoon, and although dissolved methane concentrations were roughly similar, the fluxes reached exceptional values up to about 600 mmol m$^{-2}$ d$^{-1}$. The surface heating (up to 43 °C) that strongly drives the diffusive

fluxes may contribute to these changes, but above all, the increase was consitent with the appearance of an abrupt phenomenon, the generalized ebullition of the lake.

### 4.3. O$_2$ bubble point and methane emission

A dissolved O$_2$ probe was conventionally calibrated before each measurement considering probe-atmosphere equilibrium as 100 % value. Since the atmosphere is approximately composed of one-fifth O$_2$ and four-fifths N$_2$, the value of 500 %

corresponds to O$_2$-supersaturation, i.e. the limit value of O$_2$ bubble formation in a water column in equilibrium with the atmosphere. This bubble point of O$_2$ was reached and exceeded at the beginning of the afternoon under the effect of the photosynthetic O$_2$ production in condition of strong cyanobacterial bloom and strong sunshine. Once this point was exceeded, the data showed a rapid purge of O$_2$ from the lake. The conditions of gas-supersaturation under the influence of primary production are very rare, although they are relatively standard in the green water lakes of Nhecolândia during the

dry season. In any case, to our knowledge the consequences of natural O$_2$ purging on GHG emissions have never been reported for inland waters. Such O$_2$ micro-bubbling has two effects. On the one hand, it significantly increases the surface of contact between the liquid and gaseous phase, favouring the diffusion from the liquid phase to the gaseous one inside the bubbles. On the other hand, compared to a simple diffusion process, it also increases the transfer rates of methane to the lake surface, as the ascension of the bubbles is faster than the diffusion towards the top of the water column. The microbubble

phenomenon has recently been proposed as an important flux pathway for $CH_4$ in inland aquatic systems (Beaulieu et al., 2012; McGinnis et al., 2015; Prairie and del Giorgio, 2013). Here, this process results in a $CH_4$ emission that is two orders of magnitude greater than previously reported for these environments. Surprisingly, it shows that these extreme oxic conditions that are favorable to rapid oxidation of $CH_4$ to $CO_2$ by methanotrophs correspond, on the contrary, to a hot moment of methane emission. This hot moment was maintained for 2 to 3 hours per day, during approximately 2 months at the height of

the dry season, and abruptly disappeared at the first significant rainfall.

### 4.4. Influence of early rainfall

At the first rains of the season, the bloom disappears in a few hours and massive amounts of algal organic matter are deposited at the bottom of the lake, causing a drastic change in redox and pH conditions in the upper part of the sediments. The acidification observed in the upper part of the sediments is probably due to the release of organic acids that accompany

the onset of mineralization. For a few days, these conditions become more favorable to the synthesis of $CH_4$ by methanotrophs. Then, the proximity between the sediments and the water-atmosphere exchange surface due to a thin water column (generally <0.5 m) is likely to favor methane diffusion and emissions. Unfortunately, due to the beginning of the



flooding of the Pantanal, it was not possible to continue the measures, and no value of GHG flux is associated with this period.

### 4.5. Implications for annual $CO_2$ and $CH_4$ emissions

Seasonal and daily monitoring reveals the influence of the type of biogeochemical functioning and more particularly of the bloom and its intensity on methane and carbon dioxide emissions. A quick calculation makes it possible to evaluate the consequences on annual emissions. For black-water saline lakes, emission estimates are of the order of 790 mmol $m^{-2}$ $y^{-1}$ and 73 mmol $m^{-2}$ $y^{-1}$ for $CO_2$ and $CH_4$, respectively. For the green-water saline lakes, it appears necessary to consider several situations throughout the year. On the basis of the fluxes measured outside the bloom period, the annual flux estimate revolves around 285 mmol $m^{-2}$ $y^{-1}$. This value is slightly lower, but of the same order of magnitude (about 520 mmol $m^{-2}$ $y^{-1}$) as that calculated by Bastviken et al. (2010). Taking into consideration the seasonal variations and based on 200 days without bloom throughout the rainy season, 100 days of moderate cyanobacterial bloom during the dry season, and considering that the bloom is sufficiently strong for the $O_2$–supersaturation to be reached as much as 65 days a year and for 3 hours per day, the methane flux estimate may reach 8,850 mmol $m^{-2}$ $y^{-1}$. In the latter case, no-bloom, moderate-bloom and extreme-bloom conditions represent about 2 %, 5 %, and 93 % of the yearly $CH_4$ emissions, respectively. This fast calculation highlights the significance of $O_2$ micro-bubbles on the annual methane emission, a process not considered in conventional Fickian diffusion calculations (McGinnis et al., 2015). Similarly, an estimate of the $CO_2$ consumption from green water lakes is about 1,140 mmol $m^{-2}$ $y^{-1}$, distributed in 28 %, 10 %, and 62 % during no-, moderate- and extreme-bloom conditions, respectively. Previous studies have revealed that methane emission variability within a lake may be equal to or more important than between lakes, mainly because of the diversity of habitats within lakes (Bastviken et al., 2010). Our results complement these studies by showing that, even for relatively homogeneous alkaline lakes, there is a very high seasonal variability that must be taken into account to estimate GHG emissions.

### 5 Future Studies

Nhecolândia is a vast sub-region of the Pantanal wetland characterized by a high range of salinity and pH of surface waters. The result is a diversity of environmental conditions, a major obstacle to estimating the regional contribution in terms of GHG emissions. Conducted on a limited number of saline alkaline lakes, our study clearly shows the opposition between black-water lakes, which are $CO_2$ and $CH_4$ sources with steady fluxes throughout the year, and green-water lakes characterized by a seasonal cyanobacterial bloom and behaving as $CO_2$ sink and $CH_4$ source. Although incomplete, the preliminary measurements make it possible to show the day-night flux variations, but above all the seasonal variations, which depend on the intensity of the bloom and of the $O_2$-supersaturation induced by the photosynthesis. Future measurements with floating chambers should be directed to better constrain the fluxes around these hot moments, but also immediately after the first rain and the disappearance of the bloom, since both, the decrease in pH on the surfaces of the



sediments and the low water column seems more favorable to methane production and emission. In order to achieve a good assessment of surface water emissions at the regional level, future studies should also focus on remote sensing tools to distinguish between freshwater lakes and alkaline saline lakes, but also between black and green water alkaline lakes, and ultimately allow for cyanobacterial bloom monitoring throughout the dry season.

**Acknowledgments**

This research was funded by the São Paulo Research Foundation (FAPESP n°2013/09192-0 and 2016/14227-5) by the Federal University of South Mato-Grosso (PROPP n°2011/0338) and the National Council for Scientific and Technological Development (CNPq n°443030/2015-4). It was also supported by grants awarded by São Paulo University (USP), Campinas University (UNICAMP) and the French Consulate in São Paulo. We thank Embrapa-Pantanal and Centenario farms for
providing access to the sites. J. Hesson revised the English (AcademicEnglishSolutions.com).

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



**Table 1: Date, location and general conditions during greenhouse gas emission monitoring.**

| Year | Month | Type of lake (name) | Weather conditions | Bloom conditions | EC range µS.cm⁻¹ | pH range | DOC mg.L⁻¹ | Type of measurement Number of chambers |
|------|-------|---------------------|--------------------|------------------|-------------------|----------|------------|-----------------------------------------|
| 2012 | Sept. | Black (P) | Sunny | - | 1400-1599 | 8.81-8.99 | 71 | Static 3 |
|      | Sept. | Green (V) | Sunny | Moderate | 2420-2888 | 9.48-9.73 | 236 | Static 3 |
| 2013 | Aug. | Black (P) | Sunny | - | 1715-1855 | 9.21-9.33 | 187 | Static 3 |
|      | Sept. | Green (V) | Partially cloudy | Strong | 2302-2410 | 9.67-9.78 | 465 | Static 3 |
| 2014 | Dec. | Green (M) | sunny | No | 2014-2204 | 9.37-9.51 | 102 | Dynamic 6 |
| 2015 | July | Green (M) | sunny | No | 1940-2030 | 9.28-9.37 | 122 | Dynamic 3 |
|      | Sept. | Green (G) | Sunny (evening storm) | Strong | 34000-35100 | 10.25-10.44 | 626 | Static 3 |
|      | Sept. | Black (P) | Strongly rainy | - | 1382-1450 | 9.3-9.39 | 36 | Static 3 |



**Figure 1: Location of the Pantanal wetland, Nhecolândia region, Nhumirím and Centenario farms and studied lakes.**




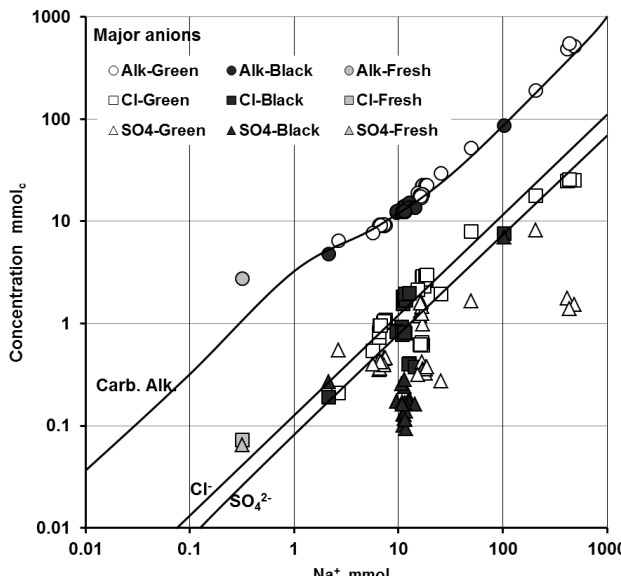
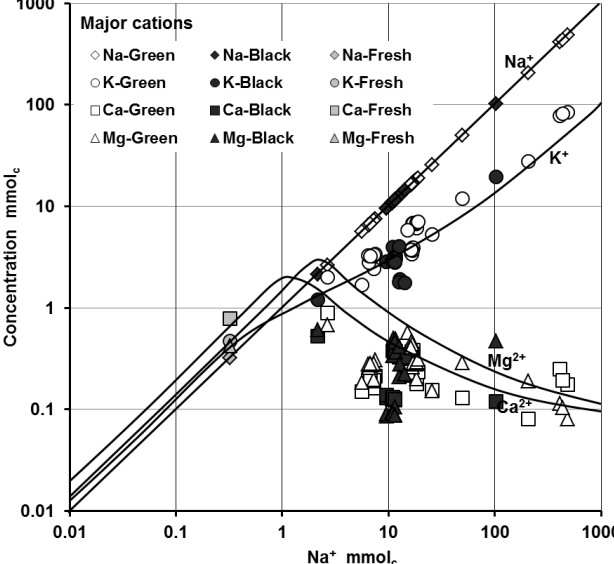

**Figure 2: Concentration diagrams based on sodium concentration showing major ions concentration of the studied lakes (fresh water lake and black- or green-water alkaline lakes). Solid lines are regional trends drawn from Furian et al. (2013).**



**Figure 3: Changes in Biogeochemical field indicators over 24 hours monitoring.**





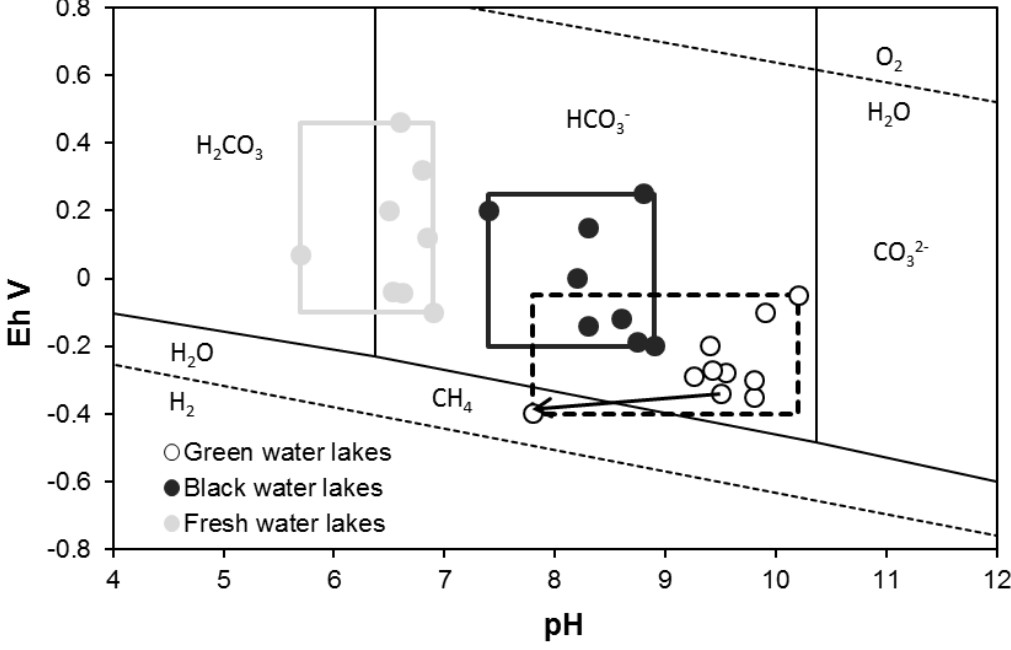

**Figure 4: Oxidation – reduction potential and pH conditions in lake sediments. Note the drop in the pH value occurring after rainfall and disappearance of the cyanobacterial bloom.**





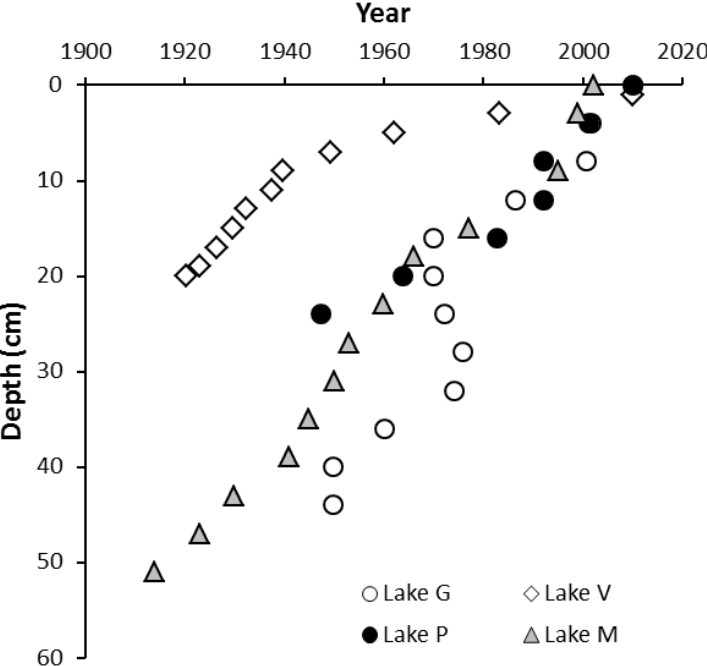

**Figure 5: Sedimentation rates obtained by unsupported $^{210}$Pb method. Results for lake M are from Fávaro et al. (2006).**



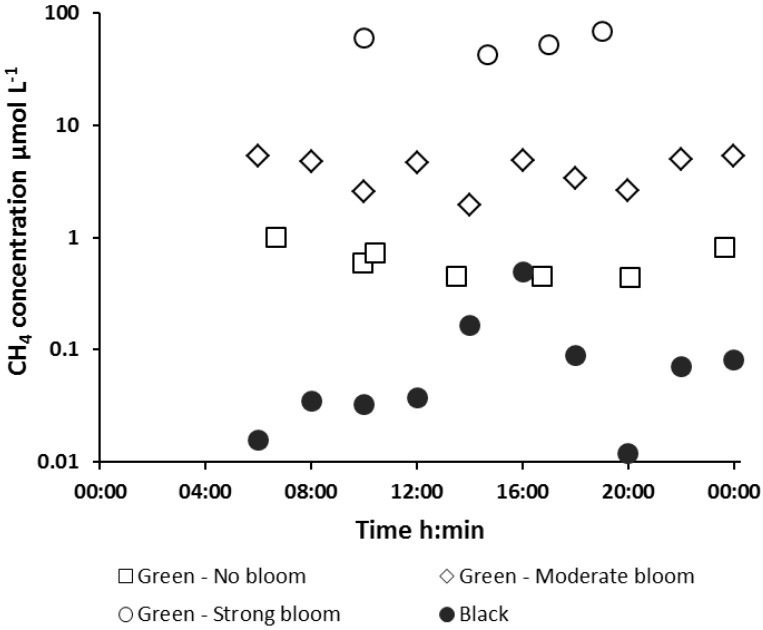

**Figure 6: Dissolved CH₄ concentrations at the top of the water column over 24 hours monitoring in black water lake P and green water lakes V and G.**




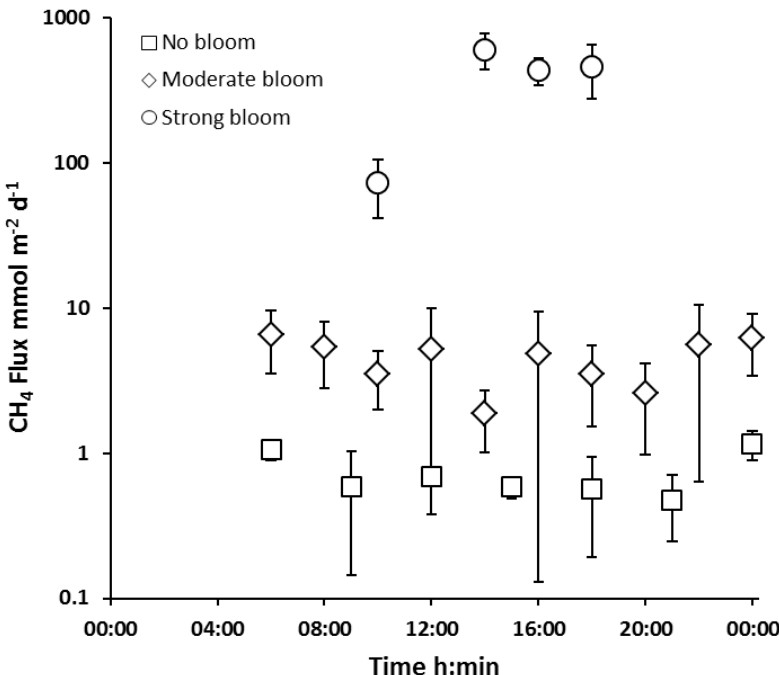

**Figure 7: Methane fluxes from green water lakes for no-, moderate- and strong bloom conditions.**




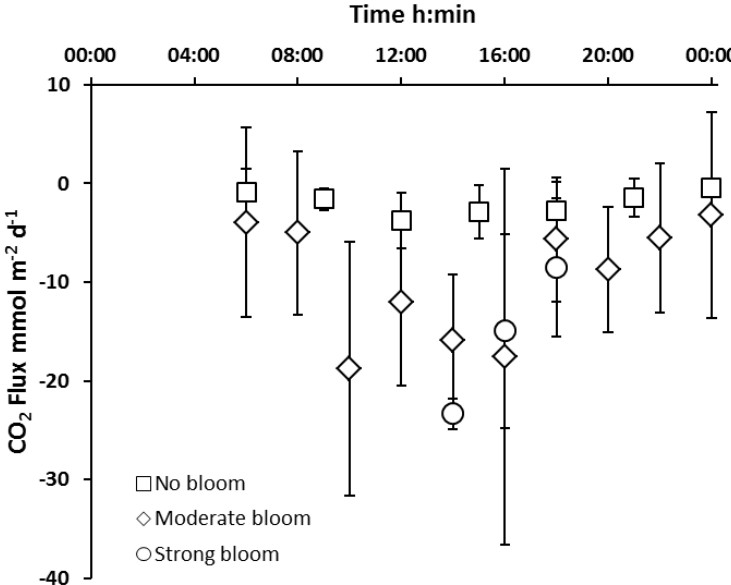

**Figure 8: Carbon dioxide fluxes from green water lakes for no-, moderate- and strong bloom conditions.**





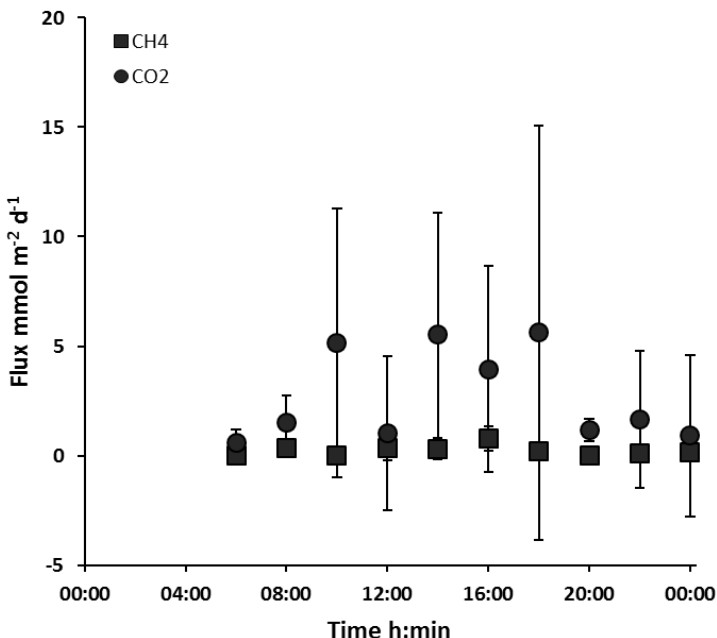

**Figure 9: Methane and carbon dioxide fluxes from black water lake P.**





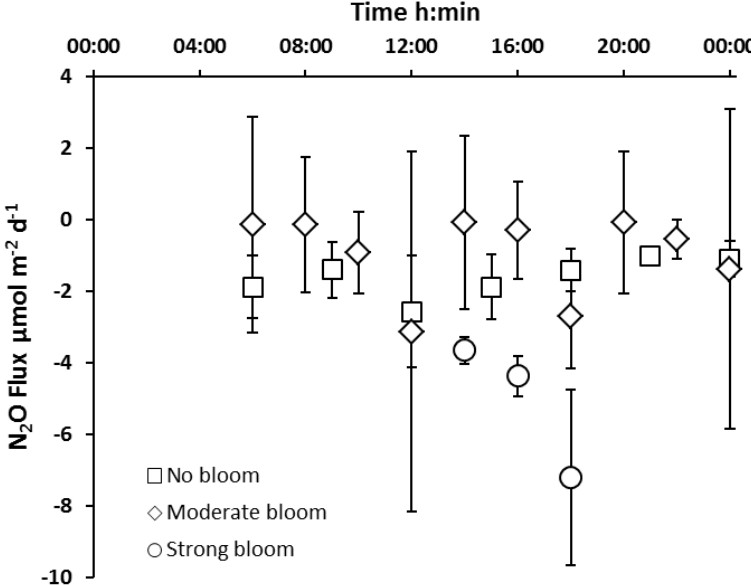

**Figure 10: Nitrous oxide fluxes from green water lakes for no-, moderate- and strong bloom conditions.**