# Peer review of "Biogeochemical diversity and hot moments of GHG emissions from shallow alkaline lakes in the Pantanal of Nhecolândia, Brazil"

_Biogeosciences, 2017_

## Referee Comment (RC1) · Anonymous Referee #1 · 20 May 2017

In theory the topic of this manuscript is good since it describes GHG fluxes in a less studies tropical wetland with considerable size. However, the paper is so poorly presented that it is even difficult to tell what has been done. Partly the problems of the paper are due to the language used and the paper will definitely benefit from language checking. The authors themselves say that this is a work providing preliminary results and that is true; the work is quite descriptive and superficial. It is unclear how this present study makes a substantial contribution in Pantanal GHG studies. The main problem is in the study design, especially the gas emission studies. After reading the manuscript several times, I still don't know how many times the gas measurements were carried out. In methods section there is no indication about dates, time etc. of

sampling. According to the figures the gas samples were taken 10 times per day on days when the systems were sampled. In Table 1 there is only sampling month, no dates. Without that information, it is impossible to judge the quality of the research. The studied systems appeared very shallow and thus they most probably are hot spots for ebullition, but ebullition was not studied at all (although it is discussed quite a lot). I find it very surprising that ebullition was ignored. From chambers the samples were drawn only at the beginning and in the end of the measuring period – which as such is strange – so the ebullition is included in the results, but in a proper study you should still measure it separately. The authors are well aware of the importance of hydrology and thus weather for their study system, but there is nothing about these basic measurements indicating that they were not monitored at all during the study period. When discussing the results, the importance of evaporation for gas fluxes is clearly stated, but despite this, heat fluxes were not measured during the study. The same applies to meteorological data in general – no measurements. There is no explanation for the selection of studies lakes, i.e. why only one freshwater lake was chosen. There is very limited background data on the lakes. It is said that the lakes are shallow, but no bathymetric maps are available. The surface area of the lakes is not presented and cannot be estimated from figure 1, since in the aerial photographs there is no scale. In general, no information about the morphometry of the lakes is available. The lakes are divided into two classes, green and black lakes, but it left unclear where the name especially of the black lakes comes from. Are they dark coloured due to DOC loading? No explanation is given for the fact that only three lakes were chosen for the sediment studies. Why these three? There are several smaller issues in methods, which require further pondering. For instance, the lakes were sampled for gas concentrations in the water, but nothing is said about the location of these sampling points and sampling depths. Temperatures were measured inside and outside of the chambers but it is not explained how these data were used. The calculations of fluxes were not explained. It is said that oxidation-reduction potential was measured also in the water (why?), not only in the sediment. These results are not shown. Nothing is said about the calibration of the fluorometer. In the results section those parts, which presumably refer to 'Biogeochemical diversity', are fairly superficial and in fact describe the basic limnology of the systems. The size of the gas bubbles is given, but it is totally unclear how the bubble studies were made. The gas emission part of results is not well structured, and needs to be rewritten to clarify the findings. In discussion gas bubbles and especially microbubbles are emphasized. However, bubbles were not studied at all and thus there is no evidence on these phenomena in the Pantanal small lakes. The advice is to be very cautious when discussing bubbles. There is also a section for the influence of rainfall. Similarly to bubbles, no information on rainfall or weather in general, so there is no proper ground for this kind of discussion. There are lots of typos and poor language. Besides gas emission part of the results section, at least the section 'studied area' should be restructured and divided at least to two paragraphs.

---

## Referee Comment (RC2) · Anonymous Referee #2 · 29 May 2017

This manuscript reports the water chemistry and greenhouse gases (GHG) emissions from lakes/ponds in the Pantanal region of Nhecolândia in Brazil. The authors found that although lakes were similar chemically (highly alkaline), they showed distinct biogeochemical functions. Black water lakes act as both CO2 and CH4 sources with low GHG emissions, while green water lakes are atmospheric CO2 sinks and CH4 source. The magnitude of the CH4 fluxes in the green water lakes depends on the presence and magnitude of the cyanobacterial blooms. The authors concluded that these lakes are active biogeochemically and may be subjected to hot moments of GHG emissions (i.e. during and after cyanobacterial blooms).

General comments

It is now known that lakes and ponds are important contributors to the global carbon cycle. They receive process and emit large amounts of $CO_2$ and $CH_4$ at rates comparable with the land and oceans. This paper thus addresses an important question regarding the biogeochemical function of lakes in a particular area, whether they act as sinks or sources of carbon to the atmosphere. The study area is of great interests by the highly spatial heterogeneity and chemical diversity of the systems and the temporal dynamics governed by floods and droughts. In general, the paper is well written and clear. I, however, have concerns about the methodology, the sampling strategy and the data presentation, which could diminish the potential impact of the paper. I also have suggestions to improve the manuscript, which I think would allow the reader to better understand the results.

Although the study area seems of great interest on its own (proven by the several studies cited by the authors on the lakes description), I think that the authors spend too much effort on the area and general lakes description, especially regarding their chemistry. Although this is interesting, it is somehow disconnected to the GHG emissions and biogeochemistry function story. The authors made very little (if no) links between the lake chemistry and GHG emission/biogeochemical function. I suggest the authors to reduce significantly the description of the lake chemistry in the region, which I think other studies did properly. If the authors still feel that the chemistry should stay as an important part of the manuscript, they should reinforce the links that exist (statistically or conceptually) between water chemistry diversity and biogeochemical functions.

Linked to the previous comment that the authors emphasized on the chemical diversity of the lakes in the study area, choosing only 6 lakes might be not representative of this highly diverse/heterogeneous region. I acknowledge however that sampling many lakes is labor (and money) intensive, and that 6 lakes are better than none. I, however, suggest the authors to rework the manuscript to better reconcile and link the great diversity of lakes to the limited sampled lakes. What makes the authors think that these lakes are representative? It seems that they are representative chemically (if I

understood right figure 2), but how are they biogeochemically? This should be clearer in the manuscript.

Specific comments

1) Lakes description:

While a thorough description of the sampled area (Pantanal lakes) was provided, the sampled lakes themselves were barely described. Basic information on lake depth, size, and thermal stratification is needed to properly interpret the chemical and biogeochemical results. Also, it is mentioned that the lakes are private and located on a farm. Are the catchments natural or managed? Forested or agriculture?

2) Data presentation:

The authors sampled the lakes during several seasons/periods (shown in Table 1) but only show 24h cycle data. Figures 3 and 6 have no error bars, while Figures 7-10 show error bars. How these means and error bars are calculated? Also, Figure 1 shows 6 lakes, but Table 1 only 4. This is very confusing for the reader. I thus strongly suggest adding a paragraph in the method section to properly describe the sampled lakes and the data used (and data not used), in which lakes, and the statistics made on the data. Also, all the figure captions should be more descriptive. For example, in Figure 3, the caption should mention what are the 4 panels, the different symbols, which day of the year it represents. . .etc.).

3) Chambers methodology: It is obvious to me that dragging chambers over the water induce artificial turbulence inside the chamber. The fluxes derived from this technique should overestimate real fluxes. The authors should provide further explanation of the potential impact of this bias on the interpretation of the results.

4) In the green water lakes, the authors estimated annual $CH_4$ flux of 8850 mmol m-2 yr-1, while $CO_2$ influx was 1140 mmol m-2 yr-1. Even if all this $CO_2$ consumption goes into biomass and that this biomass is completely used for methanogenesis, there is still

about 7500 mmol of CH4 that is missing and must be produced elsewhere. Where this methane comes from? Do the authors have ideas?

Technical corrections

P.4 L.26. I would use "shallow" instead of "low"

P11. L.16. C02 should be CO2

---

## Referee Comment (RC3) · Anonymous Referee #3 · 30 May 2017

The manuscript considers gaseous exchange with the atmosphere of shallow lakes in the Pantanal floodplain. The authors consider main findings: "Although these lakes have a similar chemical composition, the results confirm an opposition between the black-water and green-water alkaline lakes, corresponding to distinct biogeochemical functioning. Black-water lakes are $CO_2$ and $CH_4$ sources, with fairly constant emissions throughout the seasons. <...> By contrast, green-water lakes are $CO_2$ sinks but significant $CH_4$ sources with fluxes varying significantly throughout the seasons, depending on the development of the cyanobacterial bloom."

1) Some of the findings relating to gas fluxes has been already reported elsewhere, which leads this reviewer a sense of lack of novelty or insufficient search for references

by the authors. Data shown in Figures 2 and 3 present processes that are already described and well known for those lakes.

2) Methodology is very unclear.

a. How did the authors measure fluxes?

i. Static chamber: how did the authors assure that anchoring a chamber at about 10 meters apart would not disturb the nearby sediment? Have the authors carried out any experimental validation? It is unclear how gas sampling in the anchored chamber was carried out avoiding disturbing the sediments.

ii. Usually, dynamic chambers determine fluxes through on-site gas measuring system and requires the use of air pump systems. The "dynamic" approach presented by the authors is peculiar and might not negligibly disturb the water-air boundary layer. Please, verify and clarify the method. Multimedia material are welcome as pictures or video streams.

iii. How did the authors calculated and sum both fluxes by diffusion and ebullition? How many gas samples were obtained for determining a single gas flux estimate? It would be relevant to provide in the annexes the linear fits and their corresponding coefficients of determination (R2) for all measured gas fluxes.

iv. Why gas flux data are only presented hourly? Table 1 lacks information about the days of sampling; it shows only year and month.

b. All the remaining discussion depend on the quality and validity of the gas flux methodology, which is excessively dubious, and must be well clarified.

---

## Referee Comment (RC4) · Anonymous Referee #4 · 31 May 2017

Review of "Biogeochemical diversity and hot moments of GHG emissions from shallow alkaline lakes in the Pantanal of Nhecolandia, Brazil"

Summary

I'm sorry that I cannot be more encouraging at this time, and I expect that my review comments are helpful in revising your manuscript and try to improve somehow the meaning of this study. The main objective of this study (I admittedly that it is not completely clear to me) was to study the role of some physicochemical parameters in the greenhouse gas emissions-GHG ($CH_4$ and $CO_2$, I think and $N_2O$ was integrated later as complement) in shallow alkaline lakes in the Pantanal of Nhecolândia, Brazil. To this end, the authors performed several campaigns at different years at different seasons

in green and black water lakes ("freshwater lake" was omitted in the GHG results and discussion, you should remove all about this lake). I think, this manuscript contains several major weaknesses as pointed by other reviewers. For example:

Introduction

The introduction should refer more to literature of the studied type lakes (ponds) in GHG emission topics, and avoid integrate terms that are out the scope of the study and avoid unnecessary statements, for example:

(i) process-based models are mentioned but never used in the manuscript as tool, or even in the discussion of the results,

(ii) please refer properly the ideas and references in the introduction; Saunois et al. (2016) is a neat analysis of the global methane budget, but not for CO2 and N2O, then, you need to include literature about the topic (I would remove N2O, since it is a last graph with few case of the study lakes and only for "green water lakes"). There are incorrect citations for the meaning of the sentences; Bogard et al (2014) is not a good citation for the meaning of the sentence,

(iii) include more introduction about your type of greenhouse gases studied, there are few information about which gases were measured.

2 The method section is lacking some important information. To mention only a few:

(i) The most important: there is no statistical section, then, there is no idea how you determined significant differences, linear analysis, how many samples per site, time and lake were done.

(ii) You need at least a reference to indicate the advantage of the method and/or a comparison between static chamber and your namely "dynamic chamber". What is the purpose of it? Why didn't you set several static chambers and measure it? The way that you collect the gas samples is very difficult to understand. Finally, what is the purpose to use two different methods and no mentioned in the results and discussion
sections?

(iii) There is an important missing information in the gas sample procedure about the manual pressure procedure. This is a critical problem, because if you don't know the % of vacuum made, you didn't know how much dilution contain the sample injected into the vial. Did you have a pressure manometer to measure it? What is the error of the manual vacuum pump and how much volume you can extract from the vials? I said this because it represents a systematic error that may increase bias in your measurements and may be explain your error bars and not the attribution that you made to Ebullition.

(iv) The calibration for CH4 is wrong as you did it, since you used CH4 standard 10 times over to the atmospheric concentration. So, maybe you will have critical bias in the calibration curve comparing data below to 690ppm from the calibration curve.

(v) Section 2.2.2. title should be "physicochemical analysis" (or similar), since using the title "Biogeochemical field indicators" is very vague, according to the measurements made in the study.

3 Results sometimes are discussed (in the result section) in speculative way for example:

(i) I cannot see Fig 3. Opposite trend mentioned in page 35 Lines 26-29.

(ii) Carefully in the temperature results, you didn't measure at the same time (even you measured different years I think), then, temperature fluctuations is due that environmental conditions during a day, or I am wrong and you measured all lakes at the same day, did you do that?

(iii) There is no term of variation (e.g. standard deviation, standard error, variance, among others) and number of samples in the gas emission section, then I don't believe that statements as in Page 8, line 8 "The differences in the emission values between the floating chambers were moderate".

(iv) Which peaks are in Figure 7, the figure is very confusing, would be better to represent in another way (all about gas data), because error bars (also indicate what is the term of variation) are mixing and it is impossible to understand.

(v) How did you know that CH4 bubbling was moderated (Page 8 Line 12)? the heterogeneity could be as result of moving the chambers, even it is confused why you measure with different methods (static and dynamic).

(vi) Most of the time I need to assume your term of bloom and I believe you, however I cannot see the trends in the Figure 9 and why black water is not shown. What is the meaning of trend for this study? In some figures, some parameters didn't change along the time, so is it is a trend?, if I am right you need to include in Figure 9 the trend for "Black water lakes".

4 Discussion of the results remains mainly speculative, and the statements are sometimes questionable for example:

(i) Page 9 Lines 16-20 is a very vague discussion,

(ii) Please read the manuscript mentioned in Page 9 Line 29; you are working with ponds and it is out of the scope to mention methane paradox. Grossart and Tang are working with a oligotrophic lake with a very particular conditions, and Bogard et al. (2014) is an enclosure experiment to confirm methane paradox in water lakes. I think your results are more correlated to the microbial activity in sediments but no as production of methane in the water column (in oxic conditions), and your experiment doesn't allows to speculate it.

(iii) I am sorry but section 4.3 is a very speculative supposition, you didn't test any experiment to validate your supposition about microbubbling CH4. Additionally, you can't mentioned methanotrophy, since you didn't measure methanotrophy activity. Because you showed large gas emissions, it doesn't mean that methanotrophy is suppressed. You are measuring only the total emission; that is the result of the CH4 produced minus CH4 oxidized by methanotrophs. Then you don't know the rate of methanotrophy

activity, which probably is large or small, but you need to have proof to mention it.

(iv) Section 4.4 is very speculative since you didn't measure during rainfall conditions, please remove it.

―――――――――――――――――――

---

## Referee Comment (RC5) · Anonymous Referee #5 · 15 Jun 2017

GENERAL COMMENTS

It was a pleasure to review this well-conducted manuscript that adds to the growing literature on C cycling in highly dynamic tropical lakes. This manuscript entitled "Biogeochemical diversity and hot moments of GHG emissions from shallow alkaline lakes in the Pantanal of Nhecolândia, Brazil" authored by L. Barbiero et al. reveals novel findings on intense but little-known C cycling processes at low latitudes. The topic would be of high interest to readers of Biogeosciences.

However, there are serious flaws that should be considered. My main concerns are:

- The sedimentation rates are not contextualized in the introduction or objective, but

they are highlighted in the methods, results and discussion. One important point is that this ms would benefit from reporting changes in Organic Carbon Burial instead of Total Sedimentation rates. Authors should include the profile of Organic Carbon Content in the sediment at least in part of the lakes to reduce speculation in the discussion section (page 9, lines 8-20). Lakes studied here could have high organic burial even showing low total sedimentation rates and vice versa.

- Authors should reformulate the study design section to clarify differences of sampling between static and dynamic chambers. Also, they should report what lakes are assessed for each method, as there are figures with 1, 3 or 4 lakes. It's very confused in Figures 2,3, 4, 6 7 8 and 9 if lakes are different lakes or the same in different seasons.

- Overall, all legends are very poor and should be fully revised (e.g. no mention on each lake and season analyzed, number of sampling or even what means symbols and bars, such as a question: Mean and standard error?). In addition, authors should name (e.g. A, B, C. . .) panels of each figure

- The ms would benefit from any statistics treatment for Figures 6, 7, 8, 9 and 10, such as a two-way ANOVA to test the effect of different lakes and time on each key variable.

- The discussion section shows confused subsections (e.g. which were wrong like sedimentation rates within "Diversity of surface waters" or vague like "Specificities of green water alkaline lakes"). All subsection titles in the discussion section might be removed or fully revised. Authors should take care with the expressions"significant" or "significance", as they have not already addressed any statistics with their dataset.

- Also, speculative discussion on aerobic production of methane should be better addressed or removed (page 9, lines 23-31). The aquatic primary producers produce a very labile OC substrates to methanogenesis and their blooms could favor anaerobic production in the sediment, which is not necessarily oxic as waters. Indeed, few millimeters within the sediment might be enough to get anaerobic mineralization sites (see Sobek et al, 2009, Limn. & Oceanog.). Your study design does not allow interpretation

on aerobic methane production in these shallow lakes.

- Other unnecessary speculative discussion is the role of CH4 microbubbles to the total outgassing (page 10, lines 18-21). Authors should compare dissolved CH4 in surface waters with that evasion rates from chambers. They have a clear study design to confirm the role of bobbles on CH4 evasion to the atmosphere, which is not properly considered. Finally, authors should cite references to their comparisons (page 10, lines 21-22). In relation to air-water CO2 fluxes, auhors should discuss your results with the global review for alkaline lakes from Duarte et al. (2008, J. OF GEOPHYSICAL RESEARCH)

- It is not clear how could authors interpret from their results the influence on early rainfall in this subsection of discussion. The ms show same lakes before and after the rainy season. Therefore, this discussion might be possible, but the authors do not explore their results.

- In order to better address the upscaling, authors should clarify the source of the number of days without and with moderate or intense phytoplankton blooms (e.g. do they have any own dataset or only visual impression from these lakes? Or other source?). In addition, authors did not discuss any role of the observed daily variation on the upscaling.

MINOR COMMENTS

- Authors should revise the confusion related to the term "algae blooms", as other primary producer considered important (cyanobacteria) is not algae. A better term might be phytoplankton bloom. They should revise this term over the whole text.

- References are lacking in the analytical methods (e.g. page 5, section 2.2.2), and a fully revision is still needed for each method.

- Authors should include a point after the term "wetland" in the page 9 (line 11).

- The sentence "Consistent with Martins (2012). (. . .) throughout the season" (page

9, lines 7-8) should be rephrased to anything like "Our results confirmed previous evidences on the different functioning of black and green water alkaline lakes (Martins, 2012), . . ."

- What season do you mean in this complement "despite a very close mineral chemistry throughout the season" (page 9, line 8)?

- Page 10 (line 12): "strong sunshine" or "high incidence of solar radiation"?

- The terms "fast or quick calculations" over the text should be replaced to anything like "rough estimates" (e.g. page 11, lines 5 or 14-15).

- Authors should clarify that this ms assessed the variability over time and not spatially within the lake. They might complement the sentence "within the lake" to "within the lake over the daily cycle and year seasons".

- It's vague the sentence ". . . to estimate GHG emissions" (page 11, lines 20-21). What do authors mean? A regional or a global context? The GHG emissions of one of the freshwater wetland of the world? In addition, this conclusion without any argument seems speculative.

- Figure 1: Images need scale and source.

- Figure 4: What exactly means filled and dashes lines or the arrow? This kind of description should be also in the legend.

- Figures 6 and 7: Air-water fluxes and dissolved concentrations of a given gas should be in a same figure with two panels

- Figure 9: I did not understand why both CO2 and CH4 of the lake P is separated in another figure? Authors should organize all data in a same way among figures.

---

## Author Comment (AC1) · 19 Jul 2017

First, we would like to thank the 5 reviewers for the time spent on our manuscript, and for the many comments, some of which are very constructive. Some comments are redundant, as is often the case when there are many evaluators, but very few are in contradiction.

Please also note the supplement to this comment:
https://www.biogeosciences-discuss.net/bg-2017-108/bg-2017-108-AC1-supplement.pdf

[Figure]

[Figure]

**Supplement:**

Referee 1

The authors themselves say that this is a work providing preliminary results and that is true; the work is quite descriptive and superficial. It is unclear how this present study makes a substantial contribution in Pantanal GHG studies.

There is little work on GHG emissions in the Pantanal. But above all, we do not wish to consider the Pantanal as a whole, because, as mentioned in the text, it is made up of much contrasted sub-regions, mainly from a chemical point of view. We focused on Nhecolândia, the only sub-region of the Pantanal that evolves clearly in an alkaline pathway, and in which, the factor of concentration of surface waters is a major axis of environmental diversity (previous studies have shown this). Some recent studies in Nhecolândia have focused on the variability of gas emissions in space within the same lakes. These works are mentioned in our manuscript, and the annual emission budgets (in this case methane) are compared to our results. In our case, we study the variability over time, during the seasons, and based on daily cycles (when cycles have not been interrupted by weather conditions). We also emphasized the opposition between green and black-water lakes. Particularly for the most reactive green water lakes, we mention dissolved methane values of the order of 60 µmol / L. These extreme values are not common, outside the classical framework of lake studies and are worth reporting. Moreover, for similar dissolved methane values, we show that methane emissions can be multiplied by 10 because of the appearance of a new phenomenon, the exceeding of the $O_2$ bubble point, a very frequent phenomenon in some lakes of Nhecolândia. Again, it is a process known to physicists, but its environmental consequences have never been studied, and it deserves to be mentioned. In terms of contribution to GHG studies in Pantanal, the development of a regional emissions assessment will necessarily involve understanding the variability of emissions in space and time. This study contributes to this, focusing on daily and seasonal variability.

The main problem is in the study design, especially the gas emission studies. After reading the manuscript several times, I still don't know how many times the gas measurements were carried out. In methods section there is no indication about dates, time etc. of sampling.

We agree with this remark, and this aspect has been modified in the new version of the manuscript. Time of gas measurement, dates, depth of the water column (etc.) were mentioned. In addition, picture of data collection are supplied as supplement material. Below is the new text for "study design" section

"Gas fluxes from the lake to the atmosphere were measured using 32-L polyethylene floating chambers, having a base area of 0.195 m$^2$. The main conditions during the field campaigns are summarized in table 1. Two procedures were used for these measurements with fixed or slowly moving chambers. The procedure using slowly moving chambers (Photo 2 Supplement S1) was favored when the water level was sufficient and the lake diameter not too large to allow to cross from one bank to another. In this case, depending on the lake diameter, a train of 3 to 6 floating chambers was attached, leaving a gap of 10 meters between two successive floating chambers. Floating chambers were placed in the water every minute at a distance of about 30 m from the lake shore, and then slowly pulled toward the opposite bank at a maximum rate of 5 m min$^{-1}$. This experimental design allows for scanning the various water column heights, with the least turbulence disruption to the lake surface. To minimize artificial turbulence effects, foam elements were adjusted so that a maximum of 2 cm of the chamber penetrated below the water surface. The collects were carried out once each chamber reached a distance of about 30 m from the opposite bank. The collection times were variable since the first chamber reached the other margin in approximately 20 to 25 minutes, whereas the last chamber took about 35 to 40 minutes. When the water level was too low, or the lake too wide, we opted for a procedure with fixed floating chambers (Photo 3 Supplement S1). In order not to disturb the sediment just below the chamber, they were anchored with a 10-m line to avoid drifting. The line was equipped with a float to the vertical of the anchor. The chambers were located from the center to the border of the lake, and the collects were carried out after 20 min from an inflatable boat with shallow draft. Due to the low

water column, it was not possible to place a bubble shield to prevent bubbles from reaching the chamber. Therefore, the results represent the sum of both fluxes by diffusion and ebullition. For each chamber, gas samples were collected in duplicate (about 2 minutes apart) through a 60-mL syringe. Then they were transferred into 30-mL glass bottles, previously capped with gas-tight, 10-mm thick butyl rubber septa and aluminum caps, and evacuated with a hand vacuum pump at 0.75 kPa. Air samples were also collected at the departure of the chamber train for the ambient gas levels. Gas fluxes were calculated by the linear change in the amount of gas in the chambers as a function of sampled time. Thus, for example for a 6-chambers protocol, the mean and standard deviation on 12 measurements are presented as single gas emission value and error bars, respectively, for a given hour that corresponds to the launching of the first chambers. This operation was repeated every two or three hours or in order to present a complete 24-hour cycle.

Gas concentrations in the liquid phase were estimated indirectly using a headspace displacement method (Hope et al., 1995) with a 120-mL syringe and an air:water volume proportion of 1:3 (30:90 mL). For this, water samples were collected 5 cm below the surface, about 30 m from the edge of the lake. To equilibrate the headspace with the liquid phase, the syringe was shaken for 2 min by hand before injecting the headspace gas into the 30-mL glass bottle. For $CH_4$, the coefficient of gas transfer velocity ($K_{600}$, m $d^{-1}$) was calculated from the flux, the dissolved $CH_4$ concentration in water and the $CH_4$ partial pressure in the floating chamber as described by Bastviken et al. (2004)."

Below is Table 1: with the requested information

**Table 1: Date, location, lake characteristics and general conditions during greenhouse gas emission monitoring.**

| Date | Type of lake (name) Surface $km^2$ | Weather conditions | Phyt. Bloom conditions | EC range $\mu S.cm^{-1}$ | pH range | DOC $mg.L^{-1}$ | Procedure Numb of chambers | Water column range meter | Time of gas coll. Minute |
|---|---|---|---|---|---|---|---|---|---|
| Sept. 13, 2012 | Black (P) 0.087 | Sunny | - | 1400-1599 | 8.81-8.99 | 51 | Fixed 3 | 0.3 – 0.8 | 20 |
| Sept. 14, 2012 | Green (V) 0.109 | Sunny | Moderate | 2420-2888 | 9.48-9.73 | 236 | Fixed 3 | 0.1 – 0.4 | 20 |
| Aug. 30, 2013 | Black (P) 0.091 | Sunny | - | 1715-1855 | 9.21-9.33 | 37 | Fixed 3 | 0.3 – 1.1 | 20 |
| Sept. 1, 2013 | Green (V) 0.109 | Partially cloudy | Strong | 2302-2410 | 9.67-9.78 | 265 | Fixed 3 | 0.1 – 0.5 | 20 |
| Dec. 2, 2014 | Green (M) 0.053 | sunny | No | 2014-2204 | 9.37-9.51 | 102 | Sl. moving 6 | 0.1 – 0.4 | 23 to 43 |
| Jul. 7, 2015 | Green (M) 0.055 | sunny | No | 1940-2030 | 9.28-9.37 | 82 | Sl. moving 3 | 0.1 – 0.4 | 21 to 37 |
| Sept. 10, 2015 | Green (G) 0.285 | Sunny (evening storm) | Strong | 34000-35100 | 10.3-10.44 | 326 | Fixed 3 | 0.1 – 0.2 | 20 |
| Sept. 12, 2015 | Black (P) 0.093 | Strongly rainy | - | 1382-1450 | 9.3-9.4 | 36 | Fixed 3 | 0.4 – 0.7 | 20 |

And below a picture added as supplement material:

[Figure]

**Photo 2: Gas collection from a train of 6 slowly moving chambers on green water lake M in the absence of cyanobacteria bloom (December 2014). The first floating chamber has just reached the point of collection. Two samples will be collected in each chamber. The average of these 12 samples will provide 1 flux data for each gas ($CH_4$, $CO_2$ and $N_2O$).**

The studied systems appeared very shallow and thus they most probably are hot spots for ebullition, but ebullition was not studied at all (although it is discussed quite a lot). I find it very surprising that ebullition was ignored. From chambers the samples were drawn only at the beginning and in the end of the measuring period – which as such is strange – so the ebullition is included in the results, but in a proper study you should still measure it separately.

We also agree that emissions by diffusion and ebullition should be separated. Generally, this is done by installing a bubble shield, installed at a depth of about 50 cm, and which prevents the bubbles from reaching some chambers, the separation between diffusive and ebullitive flows being made by substraction between the results from the equipped and non-equipped chambers. We added a photo to illustrate the collection, and this photo emphasizes that the studied lakes are often very shallow, particularly (but not only) during low water period. The installation of such a structure below some chambers was simply impossible, as it would have disrupted the sediment, releasing gases artificially. Therefore, our results include both diffusive and ebullitive fluxes, as already mentioned in the manuscript. "*Due to the low water column, it was not possible to place a bubble shield to prevent bubbles from reaching the chamber. Therefore, the results represent the sum of both fluxes by diffusion and ebullition.*"

In addition, installing a bubble shield prevents only bubbles that come from below the shield to reach the chamber. This system is functional for bubbles emanating from the sediment. In our study, we discuss the role of ebullition from the water column (in relation to exceeding of the $O_2$ bubble point) and not from the sediment. Indeed, the microbubbles (when they begin to be visible they have a diameter of about 0.1 mm) do not emanate from the sediment. A shield would therefore give very different values depending on its depth in the water column below the chamber. It is not the information sought.

The authors are well aware of the importance of hydrology and thus weather for their study system, but there is nothing about these basic measurements indicating that they were not monitored at all during the study period.

We have a meteorological station near the sites that allowed to acquire the basic climatic data. These regional data have been complemented by local data (wind speed and air temperature) since

the temperature contrast between the lake and the surrounding forest is known to locally generate a breeze regime (Quenol et al., 2009).

When discussing the results, the importance of evaporation for gas fluxes is clearly stated, but despite this, heat fluxes were not measured during the study.

We did not mention heat flux because it is clear that the temperature increases gradually, while the increase in methane emissions is abrupt for green water lakes, and that this increase matches with (abrupt) exceeding of the O2 bubble point. It is this phenomenon that has been suspected as responsible for the increase in methane emissions. We argue with a new figure where CH4 K600 are presented, comparing lake V (strong bloom without O2 supersaturation at only 450 %, i.e. without reaching the bubble point, temp. max. about 37°C) and lake G (Strong bloom with dissolved O2 > 500 % and ebullition, temp. max. about 39°C). We mention in the discussion that "The consistent change in the calculated $K_{600}$ values (Fig. 6), which coincided with the occurrence of the abrupt generalized ebullition of lake G, emphasize that $CH_4$ behave quite differently in these 2 lakes."

[Figure]

**Figure 6: Calculated exchange gas coefficient for Methane in lakes V and G in strong bloom condition. The dashed line represents the beginning of the ebullition in lake G (13:20).**

There is no explanation for the selection of studies lakes, i.e. why only one freshwater lake was chosen. There is very limited background data on the lakes. It is said that the lakes are shallow, but no bathymetric maps are available. The surface area of the lakes is not presented and cannot be estimated from figure 1, since in the aerial photographs there is no scale. In general, no information about the morphometry of the lakes is available.

First, we can present a practical aspect to answer this comment. The entire region is made up of private properties. Research can only be done with the agreement of the owners and managers on site. This is the main constraint on the choice of lakes, and this constraint can change from year to year following the sale of a farm to a new owner that could be opposed to our work on its land. Second, before each trip, it is impossible to predict which lake will be able to bring additional and complementary data to the already existing dataset. A lake collected during a field campaign may be in the same situation during the next mission, whereas a neighboring lake may have evolved towards a stronger bloom, for example. In this case, the team will decide to collect the neighboring lake. We have focused on the acquisition of a set of data that can cover the most diverse situations.

In any case, with the restructuring of the manuscript, we have retained only 4 lakes, removing F and G for which no emission measurements had been made. The dimensions and depths are mentioned

in the text and in the table. We also mention the range of collecting depths, in the case of fixed or moving floating chambers. A scale, which was not visible in the first version, was added to Figure 1. See below the bottom part of Fig. 1.

[Figure]

Figure 1: Location of the Pantanal wetland, Nhecolândia region, Nhumirím and Centenario farms and studied lakes. Satellite images are from Google Earth™ (bar = 1 km).

The lakes are divided into two classes, green and black lakes, but it left unclear where the name especially of the black lakes comes from. Are they dark coloured due to DOC loading?

Actually both, green or black water lakes have high DOC values. These values are in the range 30 - 320 mg C / L., now mentioned in the table. Blackwater lakes have the lowest values of DOC, 30-50 mg C / L, but also show clay particles dispersed in water due to high pH and high sodium levels. For this reason, the turbidity is generally greater than 100 NTU, opaque to light and their color is distinctly dark brown. Green-water lakes have higher DOC values but no clay particles in suspension. In the absence of bloom, they are brown in color but have a lower turbidity (<20 NTU). From the beginning of the development of the bloom, the green color dominates. See also below photo 1 in supplement material S1.

[Figure]

**Photo 1: Aerial picture illustrating the contrast between a lake with black waters and a lake with green waters. Here the bloom is moderate to strong. The two lakes are about a hundred meters apart (source, matuete.com).**

No explanation is given for the fact that only three lakes were chosen for the sediment studies. Why these three?

With the restructuring of the manuscript suggested by different reviewers, we have refocused the data on 4 lakes. We provide the sedimentation data for these 4 lakes. We already had data on Lake M, we did analysis from the three other lakes. The results obtained are presented, they only allow to note, unlike expected, the absence of significant difference between the two types of lakes (Green or black water).

The lakes were sampled for gas concentrations in the water, but nothing is said about the location of these sampling points and sampling depths?

We agree with this comment, location and depth of sampling have been mentioned in the new ms: "For this, water samples were collected 5 cm below the surface, about 30 m from the edge of the lake."

Temperatures were measured inside and outside of the chambers but it is not explained how these data were used. The calculations of fluxes were not explained.

We agree, the calculation is now explained in the text. The temperature was used for the calculation of the CH4-K600 from the head space data and the CH4 concentration in the chambers.

It is said that oxidation-reduction potential was measured also in the water (why?), not only in the sediment. These results are not shown.

We agree. As geochemists, we are used to measure redox potential in water, which is giving a much wider range (and therefore wider information) than dissolved $O_2$. Anyways, these data have not been used in the study and are now removed from the "material and methods" section.

Nothing is said about the calibration of the fluorometer.

Again we agree with the comment. Anyway, the results given by the fluorometer are not relevant as for most of the case, value is over-range. It has been removed from the manuscript.

The size of the gas bubbles is given, but it is totally unclear how the bubble studies were made. The gas emission part of results is not well structured, and needs to be rewritten to clarify the findings.

The objective was not to study the bubbles, but to show that the appearance of ebullition affects the gas emissions during a few hours in the afternoon. Anyway, we agree that this part was confused and it has been re-written, with separate paragraph for CH4, CO2 and N2O, and figures have been modified accordingly grouping all information for a given gas on the same figure. See for example fig of CO2 emission below.

[Figure]

**Figure 7: Daily cycle of carbon dioxide fluxes showing emission from black water lake (P), and increasing consumption with increasing magnitude of the cyanobacterial bloom in green water lakes for no- (lake M), moderate- (lake V) and strong (lake G) bloom conditions.**

In discussion gas bubbles and especially microbubbles are emphasized. However, bubbles were not studied at all and thus there is no evidence on these phenomena in the Pantanal small lakes. The advice is to be very cautious when discussing bubbles.

A picture of the lake surface taken when the ebullition starts is added (Supplement material S1). Initially we were reluctant to introduce this illustration because the size of the bubbles on the

surface of the lake does not correspond to the size of the bubbles that form in the water column. What appears here on the picture is the grouping of hundreds of microbubbles (<0.1 mm) into larger bubbles (~1cm). These bubbles appear abruptly at the lake surface when dissolved O2 level exceeds 500% saturation, i.e. when the bubble point is reached.

[Figure]

**Photo 4: Detail of ebullition in strong bloom condition after exceeding the $O_2$ bubble point (> 500 % saturation). Microbubbles (~ 0.1 mm) do not arise from the sediment but form in the water column, and gather into larger bubbles on the surface of the lake. The abrupt emergence of ebullition, which will continue for about 3 hours during the afternoon, matches to a significant tenfold increase in methane emissions.**

There is also a section for the influence of rainfall. Similarly to bubbles, no information on rainfall or weather in general, so there is no proper ground for this kind of discussion.

This part has been shifted to the end of the discussion and in the "future direction" section. The death of the phytoplankton probably impacts gas emission, and it will be checked in the future, with other complementary data and different situations:

"Immediately after the first rain of the season, the bloom disappeared in a few hours, and substrates to methanogens, in the form of very labile organic carbon, deposited massively at the bottom of the lake. It resulted in a drastic decrease in pH in the upper part of the sediments probably due to the release of organic acids that accompany the onset of mineralization. For a few days, the drop in the sediment pH (about 2 units) created conditions more favorable to the synthesis of $CH_4$, although specific methanogens inherent to alkaline grow and produce methane at pH above 9. The proximity between the sediments and the water-atmosphere exchange surface due to a thin water column (generally <0.5 m) is likely to favor methane diffusion and emissions. Unfortunately, due to the beginning of the flooding of the Pantanal, it was not possible to continue the measures, and no value of GHG flux is associated with this period, which will have to be monitored in the future."

Besides gas emission part of the results section, at least the section 'studied area' should be restructured and divided at least to two paragraphs.

The section has been re-written and divided into 4 parts: General information on Pantanal and Nhecolândia, abiotic chemical characteristics of lakes, aspects of lake biogeochemistry, and choice of study sites.

---

## Author Comment (AC2) · 20 Jul 2017

Referee 2

I think that the authors spend too much effort on the area and general lakes description, especially regarding their chemistry. Although this is interesting, it is somehow disconnected to the GHG emissions and biogeochemistry function story. The authors made very little (if no) links between the lake chemistry and GHG emission/biogeochemical function. I suggest the authors to reduce significantly the description of the lake chemistry in the region, which I think other studies did properly.

We agree with the comment and the ms has been reorganized. Instead of providing chemistry data specific to the lakes studied here, in the form of results, we recall as a site background that lakes of nhecolândia follow a known evolution already mentioned in previous publications. This reorganization makes it possible to reduce the sections "methods", "results" and "discussion", and reinforces the "site background", also suggested by referee 1. Consequently, Fig. 2 has been removed.

Linked to the previous comment that the authors emphasized on the chemical diversity of the lakes in the study area, choosing only 6 lakes might be not representative of this highly diverse/heterogeneous region. I acknowledge however that sampling many lakes is labor (and money) intensive, and that 6 lakes are better than none. I, however, suggest the authors to rework the manuscript to better reconcile and link the great diversity of lakes to the limited sampled lakes. What makes the authors think that these lakes are representative? It seems that they are representative chemically (if I understood right figure 2), but how are they biogeochemically? This should be clearer in the manuscript.

In the new version, we mention in the "study site" section that the lakes are representative of the region from the point of view of the major elements chemistry. Only 4 lakes have been maintained for the study, 1 black-water and 3 with green-water lakes. We rely on Martins (2012) who carried out a typology of alkaline lakes, to declare that these lakes are the most representative of the alkaline lakes of the Pantanal. See below:

"The study was carried out in two different forested regions of Nhecolândia, in 3 lakes (V, P, G) at the Centenário farm (private land), and 1 lake (M) at the Nhumirím farm that belong to the Brazilian Agricultural Research Corporation (Fig. 1). The main activity of these farms is cattle breeding. The selected lakes, with a surface area between 0.05 and 0.29 $km^2$ (Table 1), are shallow with water columns hardly exceeding 1 m during the rainy season, with the exception of lake P, which can reach 2 m in its deepest point. These 4 lakes are alkaline with pH ranging from 8.9 to 10.5 throughout the year. We have previously verified that the changes in their water chemical composition are consistent with the results obtained by Furian et al. (2013) from a regional study. The alkaline lakes were selected so as to cover a wide range of water electrical conductivity, and according to the presence and magnitude of phytoplankton blooms. Lakes M, V and G are green water lakes, while P is a black water lake. These two types of lake are the most representative of the saline alkaline lakes of Nhecolândia (Martins, 2012). Because of flooding that makes it impossible to access the studied site, no data were collected in the height of the wet season, and fieldwork was concentrated in the early (May-June), medium (July-September), and in the late dry season (October-December) from 2012 to 2015. The gas emission data were acquired during 24-hour cycle monitoring (usually every 2 or 3 hours) and are supplemented by data acquired occasionally but systematically in each field campaign."

While a thorough description of the sampled area (Pantanal lakes) was provided, the sampled lakes themselves were barely described. Basic information on lake depth, size, and thermal stratification is needed to properly interpret the chemical and biogeochemical results. Also, it is mentioned that the lakes are private and located on a farm. Are the catchments natural or managed? Forested or agriculture?

Depth, size, are now given in the table, and we mentioned that the main activity of these farms is cattle breeding. It is not relevant to refer to a catchment in the flat Nhecolândia. We just maintained the sentence that was already in the first ms: "Nhecolândia has relatively closed drainage with little connection to major fluvial systems". We also mentioned in the text where the samples have been taken (distance from the lake shore, depth, etc…)

Figure 1 shows 6 lakes, but Table 1 only 4. This is very confusing for the reader. I thus strongly suggest adding a paragraph in the method section to properly describe the sampled lakes and the data used (and data not used), in which lakes, and the statistics made on the data.

We agree. As a result of the reorganization of the manuscript, lakes F and R, which were used only to strengthen the chemical framework of the sampling, were removed from the study. So there are now 4 lakes, in both the figure and the table. Sampling and statistics are better described. See below.

"Gas fluxes were calculated by the linear change in the amount of gas in the chambers as a function of sampled time. Thus, for example for a 6-chambers protocol, the mean and standard deviation on 12 measurements are presented as single gas emission value and error bars, respectively, for a given hour that corresponds to the launching of the first chambers. This operation was repeated every two or three hours or in order to present a complete 24-hour cycle."

It has been also described in the figure caption of the Supplement S1. See below.

[Figure]

**Photo 2: Gas collection from a train of 6 slowly moving chambers on green water lake M in the absence of cyanobacteria bloom (December 2014). The first floating chamber has just reached the point of collection. Two samples will be collected in each chamber. The average of these 12 samples will provide 1 flux data for each gas ($CH_4$, $CO_2$ and $N_2O$).**

And in the result section, we mentioned:
"The measurement times varying from 21 to 43 min (Table 1) on the chamber trains made it possible to verify that majority of the gas accumulation rates were tightly linear in time."

Also, all the figure captions should be more descriptive. For example, in Figure 3, the caption should mention what are the 4 panels, the different symbols, which day of the year it represents … etc.).

The figure was reviewed in accordance with the suggestions; the day of measurement is now reported in the caption and "similar climate conditions" is mentioned in the text. All the figure captions are more descriptive and informative. See below example of Fig. 2 (Former Fig. 3)

[Figure]

**Figure 2: Changes in (a) pH, (b) E.C., (c) dissolved O₂ and (d) temperature at 5 cm below the lake surface, over 24-hours monitoring. The measurement were carried out with similar climate conditions on September 13[th.], 2012 for black water lake P and green water lake G with strong bloom, and on September 14[th.], 2012 for lake V with moderate bloom. The dashed line in Fig. 2c represents the O₂ bubbling point for a solution at the equilibrium with atmospheric O₂.**

Chambers methodology: It is obvious to me that dragging chambers over the water induce artificial turbulence inside the chamber. The fluxes derived from this technique should overestimate real fluxes. The authors should provide further explanation of the potential impact of this bias on the interpretation of the results.

We understand very well the basis of this comment. As mentioned in the manuscript, this choice was motivated by the results of Bastviken (2010), so as to integrate all the heights of water column over a lake diameter. We mentioned the maximum speed of the chambers of 5 m / min. This velocity is slow but likely to create vortices in the chamber and alter water-atmosphere exchanges. However, movements of the fixed chambers were also observed under the influence of the irregular wind. Indeed, in order not to disturb the sediment just below the chamber, these chambers are anchored at a distance of 10 m, the vertical of the anchor being visualized by a float. Although anchored, these chambers can move abruptly a few m in small gusts. The bias induced by these displacements is similar. In addition, "slowly moving chambers" collection was only used twice on Lake M (see table). If the results were overestimated, values remain among the lowest in our dataset. This has no influence on the main message we want to convey, the emissions are largely changed when the O2 bubble point is exceeded, and these key data have been obtained from "fixed"

chambers, collected from inflated boat. The conditions have been more detailed in both the text and the Table. See the table below.

**Table 1: Date, location, lake characteristics and general conditions during greenhouse gas emission monitoring.**

| Date | Type of lake (name) Surface km$^2$ | Weather conditions | Phyt. Bloom conditions | EC range µS.cm$^{-1}$ | pH range | DOC mg.L$^{-1}$ | Procedure Numb of chambers | Water column range meter | Time of gas coll. Minute |
|---|---|---|---|---|---|---|---|---|---|
| Sept. 13, 2012 | Black (P) 0.087 | Sunny | - | 1400-1599 | 8.81-8.99 | 51 | Fixed 3 | 0.3 – 0.8 | 20 |
| Sept. 14, 2012 | Green (V) 0.109 | Sunny | Moderate | 2420-2888 | 9.48-9.73 | 236 | Fixed 3 | 0.1 – 0.4 | 20 |
| Aug. 30, 2013 | Black (P) 0.091 | Sunny | - | 1715-1855 | 9.21-9.33 | 37 | Fixed 3 | 0.3 – 1.1 | 20 |
| Sept. 1, 2013 | Green (V) 0.109 | Partially cloudy | Strong | 2302-2410 | 9.67-9.78 | 265 | Fixed 3 | 0.1 – 0.5 | 20 |
| Dec. 2, 2014 | Green (M) 0.053 | sunny | No | 2014-2204 | 9.37-9.51 | 102 | Sl. moving 6 | 0.1 – 0.4 | 23 to 43 |
| Jul. 7, 2015 | Green (M) 0.055 | sunny | No | 1940-2030 | 9.28-9.37 | 82 | Sl. moving 3 | 0.1 – 0.4 | 21 to 37 |
| Sept. 10, 2015 | Green (G) 0.285 | Sunny (evening storm) | Strong | 34000-35100 | 10.3-10.44 | 326 | Fixed 3 | 0.1 – 0.2 | 20 |
| Sept. 12, 2015 | Black (P) 0.093 | Strongly rainy | - | 1382-1450 | 9.3-9.4 | 36 | Fixed 3 | 0.4 – 0.7 | 20 |

4) In the green water lakes, the authors estimated annual CH4 flux of 8850 mmol m-2 yr-1, while CO2 influx was 1140 mmol m-2 yr-1. Even if all this CO2 consumption goes into biomass and that this biomass is completely used for methanogenesis, there is still about 7500 mmol of CH4 that is missing and must be produced elsewhere. Where this methane comes from? Do the authors have ideas?

Of course we agree with this constructive comment. A discussion was added on this topic: "From these data, the balance points to a carbon deficit of the order of 7500 mmol m$^{-2}$ Y$^{-1}$ in green-water lakes. This imbalance is significant, and several hypotheses must be put forward. The first point is that $CO_2$ capture was measured when the phytoplankton bloom was already very advanced. Maximum $CO_2$ capture should occur during the growth of the bloom but no data are available for this period, and the annual $CO_2$ capture budget may be underestimated. Another hypothesis is that the C budget of the lake is not balanced every year, but over the long term. A surplus year may succeed to a deficit year in terms of emission of C. This can be conceived for example in the case of a single consequent rain (> 20 mm) rather early in the dry season. This rain is likely to make the bloom disappear. It will then resume its growth but reach a sufficient density to induce $O_2$-supersaturation only close to the beginning of the rainy season when it will disappear again. Periods with $O_2$-supersaturation will be restricted, thus considerably limiting methane emissions during this year. Finally, it is possible to imagine an influx of C from the surrounding forest. However, it would be in disagreement with our observations on alkaline lakes hydrology (Furian et al., 2013; Barbiero et al., 2016). These are supplied by subsurface flows of low C-charge waters during the rainy season. On the other hand, during the dry season, transfers are made from the lake towards the beach via a soil solution with a high MO content (up to 750 mg$_C$ l$^{-1}$). This functioning would therefore tend to increase the carbon deficit in the lake."

Technical corrections

All technical corrections have been incorporated

---

## Author Comment (AC3) · 20 Jul 2017

First, we would like to thank the 5 reviewers for the time spent on our manuscript, and for the many comments, some of which are very constructive. Some comments are redundant, as is often the case when there are many evaluators, but very few are in contradiction.

Please also note the supplement to this comment:
https://www.biogeosciences-discuss.net/bg-2017-108/bg-2017-108-AC3-supplement.pdf

[Figure]

**Supplement:**

Referee 3

Some of the findings relating to gas fluxes has been already reported elsewhere, which leads this reviewer a sense of lack of novelty or insufficient search for references by the authors. Data shown in Figures 2 and 3 present processes that are already described and well known for those lakes.

We partially agree with this comment. As suggested by the other referees, the presentation of the chemistry of the studied lakes was transferred as "site background", and removed from the "results" section, the objective being no longer to show that these lakes are representative of the region, but to claim it in the presentation of the site. This reorganization of the article makes it possible to meet this demand.

On the other hand, it does not seem to us that the data presented in fig 3 are known. Notably, exceeding the $O_2$ bubble point for the green water lakes (>500% saturation) has not been described, as well as the resulting "purging" of the lake. The objective of this figure is to show the daily evolution of the parameters, and in what range.

Static chamber: how did the authors assure that anchoring a chamber at about 10 meters apart would not disturb the nearby sediment? Have the authors carried out any experimental validation? It is unclear how gas sampling in the anchored chamber was carried out avoiding disturbing the sediments.

We have specified in this version of the manuscript that the floating chambers are anchored at 10 m and that the anchorage site is equipped with a float. We assume that these precautions and this distance (10 m for a water column of less than 0.5 m) are sufficient not to disturb the sediment vertically under the chamber. We mention that the gas collects for fixed chambers were carried out from an inflatable boat.

Usually, dynamic chambers determine fluxes through on-site gas measuring system and requires the use of air pump systems. The "dynamic" approach presented by the authors is peculiar and might not negligibly disturb the water-air boundary layer. Please, verify and clarify the method. Multimedia material are welcome as pictures or video streams.

We agree with this remark, the terminology "Dynamic chambers" is not adapted and has been replaced by "slowly moving chambers". As also suggested by the other referees, a photograph was inserted to present the sampling procedure (Supplement material). Although the movement is very slow, it may induce artificial turbulence inside the chamber. In this case, the fluxes derived from this technique should overestimate real fluxes. However, "slowly moving chambers" gas collection was only used twice on Lake M (see table). If the results were overestimated, values are among the lowest in our dataset. This has no influence on the main message we want to convey, the emissions are largely changed when the bubble point is exceeded, and these key data have been obtained from "fixed" chambers.

Here is the Table

**Table 1: Date, location, lake characteristics and general conditions during greenhouse gas emission monitoring.**

| Date | Type of lake (name) Surface km² | Weather conditions | Phyt. Bloom conditions | EC range µS.cm⁻¹ | pH range | DOC mg.L⁻¹ | Procedure Numb of chambers | Water column range meter | Time of gas coll. Minute |
|---|---|---|---|---|---|---|---|---|---|
| Sept. 13, 2012 | Black (P) 0.087 | Sunny | - | 1400-1599 | 8.81-8.99 | 51 | Fixed 3 | 0.3 – 0.8 | 20 |
| Sept. 14, 2012 | Green (V) 0.109 | Sunny | Moderate | 2420-2888 | 9.48-9.73 | 236 | Fixed 3 | 0.1 – 0.4 | 20 |
| Aug. 30, 2013 | Black (P) 0.091 | Sunny | - | 1715-1855 | 9.21-9.33 | 37 | Fixed 3 | 0.3 – 1.1 | 20 |
| Sept. 1, 2013 | Green (V) 0.109 | Partially cloudy | Strong | 2302-2410 | 9.67-9.78 | 265 | Fixed 3 | 0.1 – 0.5 | 20 |
| Dec. 2, 2014 | Green (M) 0.053 | sunny | No | 2014-2204 | 9.37-9.51 | 102 | Sl. moving 6 | 0.1 – 0.4 | 23 to 43 |
| Jul. 7, 2015 | Green (M) 0.055 | sunny | No | 1940-2030 | 9.28-9.37 | 82 | Sl. moving 3 | 0.1 – 0.4 | 21 to 37 |
| Sept. 10, 2015 | Green (G) 0.285 | Sunny (evening storm) | Strong | 34000-35100 | 10.3-10.44 | 326 | Fixed 3 | 0.1 – 0.2 | 20 |
| Sept. 12, 2015 | Black (P) 0.093 | Strongly rainy | - | 1382-1450 | 9.3-9.4 | 36 | Fixed 3 | 0.4 – 0.7 | 20 |

How did the authors calculated and sum both fluxes by diffusion and ebullition? How many gas samples were obtained for determining a single gas flux estimate? It would be relevant to provide in the annexes the linear fits and their corresponding coefficients of determination ($R^2$) for all measured gas fluxes. Why gas flux data are only presented hourly? Table 1 lacks information about the days of sampling; it shows only year and month.

We acknowledge that the method of collection was not clearly presented. We have detailed: "Each chamber was collected twice, about 2 minutes apart. Thus, for example for a 6-chambers protocol, the mean and standard deviation on 12 measurements is presented as single gas emission value for a given hour corresponding to the launching of the first chambers. This operation was repeated approximately every two or three hours in order to present a complete 24-hour cycle". The day of sampling is now mentioned in the table.

The number of samples is also mentioned in the figure caption Supplement S1. See below:

[Figure]

**Photo 2: Gas collection from a train of 6 slowly moving chambers on green water lake M in the absence of cyanobacteria bloom (December 2014). The first floating chamber has just**

**reached the point of collection. Two samples will be collected in each chamber. The average of these 12 samples will provide 1 flux data for each gas ($CH_4$, $CO_2$ and $N_2O$).**

---

## Author Comment (AC4) · 20 Jul 2017

First, we would like to thank the 5 reviewers for the time spent on our manuscript, and for the many comments, some of which are very constructive. Some comments are redundant, as is often the case when there are many evaluators, but very few are in contradiction.

Please also note the supplement to this comment: https://www.biogeosciences-discuss.net/bg-2017-108/bg-2017-108-AC4-supplement.pdf

[Figure]

**Supplement:**

Referee 4

The main objective of this study (I admittedly that it is not completely clear to me) was to study the role of some physicochemical parameters in the greenhouse gas emissions-GHG (CH4 and CO2, I think and N2O was integrated later as complement) in shallow alkaline lakes in the Pantanal of Nhecolândia, Brazil.

The objective is not to test the role of some physicochemical parameters on GHG emissions. The objective is to show the daily and seasonal variability of GHG-emissions from 2 types of lakes with similar chemistry but with distinct biogeochemical functions. The introduction and objectives have been re-written. See below the final part of the introduction:

"Nhecolândia is a sub-region of the Pantanal wetland, where a myriad of shallow saline-alkaline, oligosaline and freshwater lakes and ponds coexist in the landscape, sometimes at short distances from each other (~200 m). Under the influence of cumulative evaporation over the years, the pH of some saline lakes has reached high values, close to or above 10, resulting in an increasing solubility of the organic matter, with dissolved organic carbon (DOC) values up to 750 mg L$^{-1}$ (Barbiero et al., 2016; Mariot et al., 2007). Martins (2012) noticed that waters of neighboring saline lakes with almost similar chemical composition can have permanent black or green color (Photo 1 Supplement S1), resulting from distinct biogeochemical functioning, but the parameters that control such differences are still poorly understood. Collectively, the size of the region, the number of lakes and diversity of biogeochemical conditions in space and time (day-night and seasonal change) make it difficult to estimate the regional greenhouse gas emissions from Nhecolândia. A prerequisite for such a regional balance and its contribution to the global budget is a better understanding of the diversity of scenarios and of their variability in time and space (Peixoto et al., 2015). The aim of this study is precisely to present this diversity in the specific context of Nhecolândia, and to provide preliminary results of the range of greenhouse gas fluxes (CH$_4$, CO$_2$ and N$_2$O) from the most alkaline black- and green-water lakes."

And also the contrast betwwen Black and green water lakes mentioned in the introduction:

[Figure]

**Photo 1: Aerial picture illustrating the contrast between a lake with black waters and a lake with green waters. Here the bloom is moderate to strong. The two lakes are about a hundred meters apart (source, matuete.com).**

The introduction should refer more to literature of the studied type lakes (ponds) in GHG emission topics, and avoid integrate terms that are out the scope of the study and avoid unnecessary statements, for example:

(i) process-based models are mentioned but never used in the manuscript as tool, or even in the discussion of the results,

We agree with this remark, the sentence has been modified and the reference to the process-based models has been deleted.

(ii) please refer properly the ideas and references in the introduction; Saunois et al. (2016) is a neat analysis of the global methane budget, but not for CO2 and N2O, then, you need to include literature about the topic (I would remove N2O, since it is a last graph with few case of the study lakes and only for "green water lakes").  There are incorrect citations for the meaning of the sentences; Bogard et al (2014) is not a good citation for the meaning of the sentence,

We agree. The reference to Saunois et al (2016) is maintained but related to CH4, not GHGs. Literature about CO2 and N2O was included. N2O was maintained as we are also giving emission/capture from the studied lakes. N2O emission data from black water lake, initially not shown, was added in the new manuscript. The reference to the work of Bogard et al (2014) was removed from this sentence.

(iii) include more introduction about your type of greenhouse gases studied, there are few information about which gases were measured.

OK, it has been introduce in the "objective", at the end of the introduction.  See below the first part of the Introduction section:

"Wetlands contribute to the creation of large reservoirs of biodiversity, improve the quality of surface water, reduce flood risk associated with extreme rainfall, and supply streams during low water periods (Brinson et al., 1981; Fustec and Lefeuvre, 2000; Mitsch and Gosselink, 2015; Reddy and DeLaune, 2008; Turner, 1991; Whiting and Chanton, 2001). They are also critically important to global warming because of their role in modulating atmospheric $CO_2$, $CH_4$, and $N_2O$ concentrations. (Bastviken et al., 2004, 2011; Liengaard et al., 2012; Wang et al., 1993). Among the wetlands, tropical wetlands are known to be highly reactive, as permanent high temperatures increase the velocity of the biogeochemical reactions (Fustec and Lefeuvre, 2000; Reddy and DeLaune, 2008). As an additional restriction, continental alkaline wetlands are characterized by an increase in water pH during evaporation, favoring the solubilization, transfer and accumulation of organic matter in the landscape. Collectively, these conditions can lead to highly reactive portions of landscape; i.e. emission variability in time and space where the greenhouse gas fluxes are poorly constrained (Peixoto et al., 2015).
$CH_4$ released from wetlands accounts for more than 75% of natural $CH_4$ source, and more than 20% of the global $CH_4$ source (Schlesinger, 1997), although an important uncertainty on the $CH_4$ global budget is today attributable to emissions from wetland and other inland waters (Saunois et al., 2016). A data compilation from 196 saline lakes around the world highlighted their role in the global $CO_2$ emission. Lakes with pH below 9 were identified rather as $CO_2$ sources, while the most alkaline ones, with higher primary production, were generally weak $CO_2$ sinks (Duarte et al., 2008). Regarding $N_2O$, global emission remains largely uncertain, ranging from 6.7 to 36.6 Tg N/yr (IPCC, 2007). About 25% of the global $N_2O$ emission is attributed to uncultivated tropical soils, but exact locations and controlling mechanisms are not clear. Wetland ecosystems contribute considerably to $N_2O$ budgets (XU et al., 2008) and Liengaard et al. (2012) suggest that the Pantanal wetland in Brazil potentially contributes about 1.7%, a significant single source of $N_2O$. In this context, to understand the various processes controlling inland water emissions is still regarded as a priority."

2 The method section is lacking some important information. To mention only a few:

(i) The most important: there is no statistical section, then, there is no idea how you determined significant differences, linear analysis, how many samples per site, time and lake were done.

The method was not fully detailed in the previous version; it has been re-written. In particular, we explain that each value presented hourly is the mean value over duplicate measurements in all floating chamber (3 chambers = 6 measurements or 6 chambers = 12 measurements), and that the error bars denote the standard deviation. The number of chambers is mentioned in the table. See below the table and the modified method section:

**Table 1: Date, location, lake characteristics and general conditions during greenhouse gas emission monitoring.**

| Date | Type of lake (name) Surface km$^2$ | Weather conditions | Phyt. Bloom conditions | EC range µS.cm$^{-1}$ | pH range | DOC mg.L$^{-1}$ | Procedure Numb of chambers | Water column range meter | Time of gas coll. Minute |
|---|---|---|---|---|---|---|---|---|---|
| Sept. 13, 2012 | Black (P) 0.087 | Sunny | - | 1400-1599 | 8.81-8.99 | 51 | Fixed 3 | 0.3 – 0.8 | 20 |
| Sept. 14, 2012 | Green (V) 0.109 | Sunny | Moderate | 2420-2888 | 9.48-9.73 | 236 | Fixed 3 | 0.1 – 0.4 | 20 |
| Aug. 30, 2013 | Black (P) 0.091 | Sunny | - | 1715-1855 | 9.21-9.33 | 37 | Fixed 3 | 0.3 – 1.1 | 20 |
| Sept. 1, 2013 | Green (V) 0.109 | Partially cloudy | Strong | 2302-2410 | 9.67-9.78 | 265 | Fixed 3 | 0.1 – 0.5 | 20 |
| Dec. 2, 2014 | Green (M) 0.053 | sunny | No | 2014-2204 | 9.37-9.51 | 102 | Sl. moving 6 | 0.1 – 0.4 | 23 to 43 |
| Jul. 7, 2015 | Green (M) 0.055 | sunny | No | 1940-2030 | 9.28-9.37 | 82 | Sl. moving 3 | 0.1 – 0.4 | 21 to 37 |
| Sept. 10, 2015 | Green (G) 0.285 | Sunny (evening storm) | Strong | 34000-35100 | 10.3-10.44 | 326 | Fixed 3 | 0.1 – 0.2 | 20 |
| Sept. 12, 2015 | Black (P) 0.093 | Strongly rainy | - | 1382-1450 | 9.3-9.4 | 36 | Fixed 3 | 0.4 – 0.7 | 20 |

"Gas fluxes from the lake to the atmosphere were measured using 32-L polyethylene floating chambers, having a base area of 0.195 m$^2$. The main conditions during the field campaigns are summarized in table 1. Two procedures were used for these measurements with fixed or slowly moving chambers. The procedure using slowly moving chambers (Photo 2 Supplement S1) was favored when the water level was sufficient and the lake diameter not too large to allow to cross from one bank to another. In this case, depending on the lake diameter, a train of 3 to 6 floating chambers was attached, leaving a gap of 10 meters between two successive floating chambers. Floating chambers were placed in the water every minute at a distance of about 30 m from the lake shore, and then slowly pulled toward the opposite bank at a maximum rate of 5 m min$^{-1}$. This experimental design allows for scanning the various water column heights, with the least turbulence disruption to the lake surface. To minimize artificial turbulence effects, foam elements were adjusted so that a maximum of 2 cm of the chamber penetrated below the water surface. The collects were carried out once each chamber reached a distance of about 30 m from the opposite bank. The collection times were variable since the first chamber reached the other margin in approximately 20 to 25 minutes, whereas the last chamber took about 35 to 40 minutes. When the water level was too low, or the lake too wide, we opted for a procedure with fixed floating chambers (Photo 3 Supplement S1). In order not to disturb the sediment just below the chamber, they were anchored with a 10-m line to avoid drifting. The line was equipped with a float to the vertical of the anchor. The chambers were located from the center to the border of the lake, and the collects were carried out after 20 min from an inflatable boat with shallow draft. Due to the low water column, it was not possible to place a bubble shield to prevent bubbles from reaching the chamber. Therefore, the results represent the sum of both fluxes by diffusion and ebullition. For each chamber, gas samples were collected in duplicate (about 2 minutes apart) through a 60-mL syringe. Then they were transferred into 30-mL glass bottles, previously capped with gas-tight, 10-mm thick butyl rubber septa and aluminum caps, and evacuated with a hand vacuum pump at 0.75 kPa. Air samples were also collected at the departure of the chamber train for the ambient gas levels. Gas fluxes were calculated by the linear change in the amount of gas in the chambers as a function of sampled time. Thus, for example for a 6-chambers protocol, the mean and standard

deviation on 12 measurements are presented as single gas emission value and error bars, respectively, for a given hour that corresponds to the launching of the first chambers. This operation was repeated every two or three hours or in order to present a complete 24-hour cycle."

And here is the Photo 2 Supplement mentioned in the text:

[Figure]

**Photo 2: Gas collection from a train of 6 slowly moving chambers on green water lake M in the absence of cyanobacteria bloom (December 2014). The first floating chamber has just reached the point of collection. Two samples will be collected in each chamber. The average of these 12 samples will provide 1 flux data for each gas ($CH_4$, $CO_2$ and $N_2O$).**

(ii) You need at least a reference to indicate the advantage of the method and/or a comparison between static chamber and your namely "dynamic chamber".   What is the purpose of it?  Why didn't you set several static chambers and measure it?  The way that you collect the gas samples is very difficult to understand. Finally, what is the purpose to use two different methods and no mentioned in the results and discussion sections?

The use of fixed floating chambers provides information on emissions at a fixed point, with a given water column height. It also requires various shifts of the collection equipment. In the case of slowly moving chambers, the chambers scan and involve all the lake water column heights covering the length of a diameter. All the collections are made in series, at the same point, which avoids having to move the equipment. Unfortunately, this procedure was not applicable for larger lakes, or when portions of the lake had a column of water that was too shallow. A photo is provided in "supplement material", allowing to visualize the "slowly moving" procedure. See above.

(iii) There is an important missing information in the gas sample procedure about the manual pressure procedure. This is a critical problem, because if you don't know the % of vacuum made, you didn't know how much dilution contain the sample injected into the vial. Did you have a pressure manometer to measure it?

We agree with this comments. The vacuum pump used was indeed fitted with a manometer. The depression obtained is now mentioned in the text (0.75 KPa). See above.

(iv) The calibration for CH4 is wrong as you did it, since you used CH4 standard 10 times over to the atmospheric concentration.  So, maybe you will have critical bias in the calibration curve comparing data below to 690ppm from the calibration curve.

We agree. The value mentioned in the first ms was wrong. This part of the method has been actualized as follow: "Gas concentrations ($CH_4$, $CO_2$ and $N_2O$) were measured by gas chromatography model Shimadzu GC-2014 (Shimadzu Co., Columbia, MD, 5 USA). The chromatographer was equipped with a packed column, an electron capture detector (ECD) to analyze $N_2O$, and a flame ionization detector (FID) to quantify $CO_2$ and $CH_4$. Prior to detection, $CO_2$ was reduced to $CH_4$ using a methanizer. The gas analyzer was calibrated with NOAA CMDL certified standars $CO_2$ (357.5 and 1531 ppm), $CH_4$ (1.016 and 9.639 ppm) and $N_2O$ (313 and 11,240 ppb) gas standards (minimum and maximum, respectively). Analytical accuracy was better than 0.02 ppm $CH_4$ and precision was better than 0.005 ppm expressed as the standard error of the mean for multiple measurements of standards. The analyses were performed in the Environmental Science Laboratory (UFSCar, Sorocaba, Brazil)."

(v) Section 2.2.2. title should be "physicochemical analysis" (or similar), since using the title "Biogeochemical field indicators" is very vague, according to the measurements made in the study.

Of course, it has been changed to "Field physico-chemical measurements".

3 Results sometimes are discussed (in the result section) in speculative way for example:

(i) I cannot see Fig 3. Opposite trend mentioned in page 35 Lines 26-29.

"Opposite" has been changed to "distinct" and "opposition" to "difference".

(ii) Carefully in the temperature results, you didn't measure at the same time (even you measured different years I think), then, temperature fluctuations is due that environmental conditions during a day, or I am wrong and you measured all lakes at the same day, did you do that?

We agree. The date corresponding to the measurements given as an example in Fig. 3 is mentioned. For Lakes P and G, measurements were taken on the same day, the two lakes being only a hundred meters apart. For Lake V, the data corresponds to the next day, and we specify that the meteorological conditions were similar. See below:

[Figure]

**Figure 2: Changes in (a) pH, (b) E.C., (c) dissolved O₂ and (d) temperature at 5 cm below the lake surface, over 24-hours monitoring. The measurement were carried out with similar climate conditions on September 13th., 2012 for black water lake P and green water lake G with strong bloom, and on September 14th., 2012 for lake V with moderate bloom. The dashed line in Fig. 2c represents the O₂ bubbling point for a solution at the equilibrium with atmospheric O₂.**

(iii) There is no term of variation (e.g. standard deviation, standard error, variance, among others) and number of samples in the gas emission section, then I don't believe that statements as in Page 8, line 8 "The differences in the emission values between the floating chambers were moderate".

This part has been modified and detailed. In particular, we detailed the achievement of results at each hour of collection. "Gas fluxes were calculated by the linear change in the amount of gas in the chambers as a function of sampled time. Thus, for example for a 6-chambers protocol, the mean and standard deviation on 12 measurements is presented as single gas emission value and error bars, respectively, for a given hour that corresponds to the launching of the first chambers. This operation was repeated every two or three hours, in order to present a complete 24-hour cycle."

(iv) Which peaks are in Figure 7, the figure is very confusing, would be better to represent in another way (all about gas data), because error bars (also indicate what is the term of variation) are mixing and it is impossible to understand.

We agree with this comment. As was also suggested by other referees, we have grouped all the data of a given gas on the same graph. To reduce the confusion in the error bars, we opted for different colors (dark brown for black water lakes, and different shades of green for green water lakes). See

below, for example, the figure for methane emission (dissolved methane and fluxes are grouped on a single Fig):

[Figure]

**Figure 5: (a) Dissolved methane concentrations at the top of the water column, (b) and methane fluxes over 24 hours monitoring in black water lake (lake P) and green water lakes for no- (lake M), moderate- (lake V) and strong- (lake G) bloom conditions. Due to the logarithmic scale used, some negative values of the error bars (denoting standard deviations) are not drawn. The dashed line represents the beginning of the ebullition in lake G (13:20).**

(v) How did you know that CH4 bubbling was moderated (Page 8 Line 12)? the heterogeneity could be as result of moving the chambers, even it is confused why you measure with different methods (static and dynamic).

In fact, as has been pointed out by other referees, nothing allows to discuss on the role of methane ebullition from the sediment. This part has been deleted from the manuscript.

(vi) Most of the time I need to assume your term of bloom and I believe you, however I cannot see the trends in the Figure 9 and why black water is not shown. What is the meaning of trend for this study? In some figures, some parameters didn't change along the time, so is it is a trend?, if I am right you need to include in Figure 9 the trend for "Black water lakes".

A picture showing the contrast between two lakes with black and green waters is proposed as supplement material. In this picture, the intensity of the bloom is obvious, and here we have only a moderate to strong bloom, not enough for the O2 bubble point to be exceeded in the afternoon. "Trend" was removed from the sentence. However we maintained it for black water lakes: "For the black water lake, no clear trend towards emission or consumption of N2O was observed." These N2O data on black water lake have been incorporated in the figure. See above and below:

[Figure]

**Figure 8: Nitrous oxide fluxes from black water lake and green water lakes for no-, moderate- and strong bloom conditions.**

4 Discussion of the results remains mainly speculative, and the statements are sometimes questionable for example:

(i) Page 9 Lines 16-20 is a very vague discussion,

This is actually not a discussion, but a link to discussion furtherly developed in the ms.

(ii) Please read the manuscript mentioned in Page 9 Line 29; you are working with ponds and it is out of the scope to mention methane paradox. Grossart and Tang are working with a oligotrophic lake with a very particular conditions, and Bogard et al. (2014) is an enclosure experiment to confirm methane paradox in water lakes. I think your results are more correlated to the microbial activity in sediments but no as production of methane in the water column (in oxic conditions), and your experiment doesn't allows to speculate it.

We agree, this speculative discussion was removed from the manuscript.

(iii) I am sorry but section 4.3 is a very speculative supposition, you didn't test any experiment to validate your supposition about microbubbling CH4. Additionally, you can't mentioned methanotrophy, since you didn't measure methanotrophy activity. Because you showed large gas emissions, it doesn't mean that methanotrophy is suppressed. You are measuring only the total emission; that is the result of the CH4 produced minus CH4 oxidized by methanotrophs. Then you don't know the rate of methanotrophy activity, which probably is large or small, but you need to have proof to mention it.

As suggested by other referees this section has been re-written. We focused on the O2 microbubbling, showing that the onset of this phenomenon enhanced methane emissions. A figure

was added comparing the calculated K600 coefficient for lakes V and G. "The consistent change in the calculated $K_{600}$ values (Fig. 6), which coincided with the occurrence of the abrupt generalized ebullition of lake G, emphasize that $CH_4$ behave quite differently in these 2 lakes."

With regard to methanotrophy, we agree with the comment. Of course, nothing in our dataset allows us to discuss methanotrophy. The error originally comes from a confusion methanotrophic / methanogenic. This point has been changed.

[Figure]

**Figure 6: Calculated exchange gas coefficient for Methane in lakes V and G in strong bloom condition. The dashed line represents the beginning of the ebullition in lake G (13:20).**

(iv) Section 4.4 is very speculative since you didn't measure during rainfall conditions, please remove it.

This part was removed from the discussion and shifted to the section "future directions"

---

## Author Comment (AC5) · 20 Jul 2017

Referee 5

- The sedimentation rates are not contextualized in the introduction or objective, but they are highlighted in the methods, results and discussion. One important point is that this ms would benefit from reporting changes in Organic Carbon Burial instead of Total Sedimentation rates. Authors should include the profile of Organic Carbon Content in the sediment at least in part of the lakes to reduce speculation in the discussion section (page 9, lines 8-20). Lakes studied here could have high organic burial even showing low total sedimentation rates and vice versa.

We fully agree with this comment. Unfortunately, our organic carbon data set is not complete (or not fully reliable). Therefore, we decided to present only the total sedimentation rates. The data will be crossed with organic carbon data in the future (however, not in this paper) to present the organic carbon burial.

- Authors should reformulate the study design section to clarify differences of sampling between static and dynamic chambers. Also, they should report what lakes are assessed for each method, as there are figures with 1, 3 or 4 lakes. It's very confused in Figures 2,3, 4, 6 7 8 and 9 if lakes are different lakes or the same in different seasons.

This study design section has been thoroughly reformulated and detailed. The information on the two procedures, the conditions that led to the choice of each procedure, date, number of samples, water column depth, collection times, size of the lakes, (etc.) are now described in the text and in the table. To this, we added 2 pictures of the collection procedure in supplement S1. We cannot show all the collects. We selected some daily cycles representative of what happened in the lakes depending on the conditions. In Fig 2, we have chosen different lakes but on a given date, in order to allow to ignore variations of weather conditions. See below the main changes:

Study design section:
"Gas fluxes from the lake to the atmosphere were measured using 32-L polyethylene floating chambers, having a base area of 0.195 m$^2$. The main conditions during the field campaigns are summarized in table 1. Two procedures were used for these measurements with fixed or slowly moving chambers. The procedure using slowly moving chambers (Photo 2 Supplement S1) was favored when the water level was sufficient and the lake diameter not too large to allow to cross from one bank to another. In this case, depending on the lake diameter, a train of 3 to 6 floating chambers was attached, leaving a gap of 10 meters between two successive floating chambers. Floating chambers were placed in the water every minute at a distance of about 30 m from the lake shore, and then slowly pulled toward the opposite bank at a maximum rate of 5 m min$^{-1}$. This experimental design allows for scanning the various water column heights, with the least turbulence disruption to the lake surface. To minimize artificial turbulence effects, foam elements were adjusted so that a maximum of 2 cm of the chamber penetrated below the water surface. The collects were carried out once each chamber reached a distance of about 30 m from the opposite bank. The collection times were variable since the first chamber reached the other margin in approximately 20 to 25 minutes, whereas the last chamber took about 35 to 40 minutes. When the water level was too low, or the lake too wide, we opted for a procedure with fixed floating chambers (Photo 3 Supplement S1). In order not to disturb the sediment just below the chamber, they were anchored with a 10-m line to avoid drifting. The line was equipped with a float to the vertical of the anchor. The chambers were located from the center to the border of the lake, and the collects were carried out after 20 min from an inflatable boat with shallow draft. Due to the low water column, it was not possible to place a bubble shield to prevent bubbles from reaching the chamber. Therefore, the results represent the sum of both fluxes by diffusion and ebullition. For each chamber, gas samples were collected in duplicate (about 2 minutes apart) through a 60-mL syringe. Then they were transferred into 30-mL glass bottles, previously capped with gas-tight, 10-mm thick butyl rubber septa and aluminum caps, and evacuated with a hand vacuum pump at 0.75 kPa. Air samples were also collected at the departure of the chamber train for the ambient gas

levels. Gas fluxes were calculated by the linear change in the amount of gas in the chambers as a function of sampled time. Thus, for example for a 6-chambers protocol, the mean and standard deviation on 12 measurements are presented as single gas emission value and error bars, respectively, for a given hour that corresponds to the launching of the first chambers. This operation was repeated every two or three hours or in order to present a complete 24-hour cycle."

Table:

**Table 1: Date, location, lake characteristics and general conditions during greenhouse gas emission monitoring.**

| Date | Type of lake (name) Surface km² | Weather conditions | Phyt. Bloom conditions | EC range µS.cm⁻¹ | pH range | DOC mg.L⁻¹ | Procedure Numb of chambers | Water column range meter | Time of gas coll. Minute |
|---|---|---|---|---|---|---|---|---|---|
| Sept. 13, 2012 | Black (P) 0.087 | Sunny | - | 1400-1599 | 8.81-8.99 | 51 | Fixed 3 | 0.3 – 0.8 | 20 |
| Sept. 14, 2012 | Green (V) 0.109 | Sunny | Moderate | 2420-2888 | 9.48-9.73 | 236 | Fixed 3 | 0.1 – 0.4 | 20 |
| Aug. 30, 2013 | Black (P) 0.091 | Sunny | - | 1715-1855 | 9.21-9.33 | 37 | Fixed 3 | 0.3 – 1.1 | 20 |
| Sept. 1, 2013 | Green (V) 0.109 | Partially cloudy | Strong | 2302-2410 | 9.67-9.78 | 265 | Fixed 3 | 0.1 – 0.5 | 20 |
| Dec. 2, 2014 | Green (M) 0.053 | sunny | No | 2014-2204 | 9.37-9.51 | 102 | Sl. moving 6 | 0.1 – 0.4 | 23 to 43 |
| Jul. 7, 2015 | Green (M) 0.055 | sunny | No | 1940-2030 | 9.28-9.37 | 82 | Sl. moving 3 | 0.1 – 0.4 | 21 to 37 |
| Sept. 10, 2015 | Green (G) 0.285 | Sunny (evening storm) | Strong | 34000-35100 | 10.3-10.44 | 326 | Fixed 3 | 0.1 – 0.2 | 20 |
| Sept. 12, 2015 | Black (P) 0.093 | Strongly rainy | - | 1382-1450 | 9.3-9.4 | 36 | Fixed 3 | 0.4 – 0.7 | 20 |

And Supplements:

[Figure]

**Photo 2: Gas collection from a train of 6 slowly moving chambers on green water lake M in the absence of cyanobacteria bloom (December 2014). The first floating chamber has just reached the point of collection. Two samples will be collected in each chamber. The average of these 12 samples will provide 1 flux data for each gas (CH₄, CO₂ and N₂O).**

[Figure]

**Photo 3: Gas collection from a set of 3 fixed floating chambers on Lake G with strong bloom condition (September 10, 2015). Two samples will be collected in each chamber after 20 min. The average of these 6 samples will provide 1 flux data for each gas (CH₄, CO₂ and N₂O). Note that the ebullition of the lake due to O₂-supresaturation has started.**

*- Overall, all legends are very poor and should be fully revised (e.g. no mention on each lake and season analyzed, number of sampling or even what means symbols and bars, such as a question: Mean and standard error?). In addition, authors should name (e.g. A, B, C...) panels of each figure.*

We agree. The captions have been strengthened. The lake is now mentioned in the caption and the date in the Table. The number of samples is given in the table, and the meaning of the error bars is specified in the text. The figures are grouped into different panels and we opted for colored figures in order to reduce the confusion associated with the superposition of the error bars. See table above, and for example Fig. 5 and 7 below:

[Figure]

**Figure 5: (a) Dissolved methane concentrations at the top of the water column, (b) and methane fluxes over 24 hours monitoring in black water lake (lake P) and green water lakes for no- (lake M), moderate- (lake V) and strong- (lake G) bloom conditions. Due to the logarithmic scale used, some negative values of the error bars (denoting standard deviations) are not drawn. The dashed line represents the beginning of the ebullition in lake G (13:20).**

[Figure]

**Figure 7: Daily cycle of carbon dioxide fluxes showing emission from black water lake (P), and increasing consumption with increasing magnitude of the cyanobacterial bloom in green water lakes for no- (lake M), moderate- (lake V) and strong (lake G) bloom conditions.**

- The ms would benefit from any statistics treatment for Figures 6, 7, 8, 9 and 10, such as a two-way ANOVA to test the effect of different lakes and time on each key variable.

We agree with the suggestion, but we decided not to introduce ANOVA treatment at this stage of data publication on this lake system. The data set is still too thin. But we have received funding to continue this research on other types of lakes (lakes with red waters, crystalline waters or non-alkaline lakes). This type of statistical processing will be applied to the complete data set at the end of the project.

- The discussion section shows confused subsections (e.g. which were wrong like sedimentation rates within "Diversity of surface waters" or vague like "Specificities of green water alkaline lakes"). All subsection titles in the discussion section might be removed or fully revised. Authors should take care with the expressions "significant" or "significance", as they have not already addressed any statistics with their dataset.

We agree and decided to remove all subsection titles. "Significance" and "significant" were changed for "importance" and "important".

- Also, speculative discussion on aerobic production of methane should be better addressed or removed (page 9, lines 23-31). The aquatic primary producers produce a very labile OC substrates to methanogenesis and their blooms could favor anaerobic production in the sediment, which is not necessarily oxic as waters. Indeed, few millimeters within the sediment might be enough to get anaerobic

mineralization sites (see Sobek et al, 2009, Limn. & Oceanog.). Your study design does not allow interpretation on aerobic methane production in these shallow lakes.

We agree, this speculative discussion was removed from the manuscript. See also reply to other referees.

- Other unnecessary speculative discussion is the role of CH4 microbubbles to the total outgassing (page 10, lines 18-21). Authors should compare dissolved CH4 in surface waters with that evasion rates from chambers. They have a clear study design to confirm the role of bobbles on CH4 evasion to the atmosphere, which is not properly considered. Finally, authors should cite references to their comparisons (page 10, lines 21-22). In relation to air-water CO2 fluxes, auhors should discuss your results with the global review for alkaline lakes from Duarte et al. (2008, J. OF GEOPHYSICAL RESEARCH)

We agree. In the new manuscript, we focused on the differences between lake V and G (both with strong bloom development) to highlight that CH4 behaves quite differently on these two lakes. In particular, we introduced the calculation of the CH4-K600 that shifted from about 1.3 to above 4 when O2-bubbling started. In the discussion, our data are compared with the results obtained by Duarte et al. (2008). See below:

[Figure]

**Figure 6: Calculated exchange gas coefficient for Methane in lakes V and G in strong bloom condition. The dashed line represents the beginning of the ebullition in lake G (13:20).**

Discussion section:

"The consistent change in the calculated $K_{600}$ values (Fig. 6), which coincided with the occurrence of the abrupt generalized ebullition of lake G, emphasize that $CH_4$ behave quite differently in these 2 lakes."

And also:

"A rough estimate makes it possible to evaluate the consequences on annual emissions. For black-water alkaline lakes, emission estimates are of the order of 790 mmol m$^{-2}$ y$^{-1}$ and 73 mmol m$^{-2}$ y$^{-1}$

for $CO_2$ and $CH_4$, respectively. In agreement with the observations of Duarte et al. (2008) from global review for saline lakes, black-water alkaline lakes of Nhecolândia are closer to a group of saline lakes with pH below 9, which are generally stronger sources of $CO_2$ to the atmosphere. However, in our case their contribution appears much lower than the mean value calculated by these authors (2.16 against 81-105 mmol m$^{-2}$ d$^{-1}$).

By contrast, green-water lakes behave similarly to saline alkaline lakes with pH greater than 9, which are more productive and consequently have lower $CO_2$ partial pressure, and are commonly weak $CO_2$ sinks. For these green-water saline lakes, it appears necessary to consider several situations throughout the year. On the basis of the fluxes measured outside the bloom period, the annual $CH_4$ flux estimate revolves around 285 mmol m$^{-2}$ y$^{-1}$. This value is slightly lower, but of the same order of magnitude (about 520 mmol m$^{-2}$ y$^{-1}$) as that calculated by Bastviken et al. (2010). Based on 4 years of observation (2012-2015), a year can be divided into approximately 200 days without bloom throughout the rainy season, 100 days with moderate phytoplankton bloom during the dry season, and 65 days with a bloom magnitude sufficient for the $O_2$–supersaturation to be reached for 3 hours per day. Taking into consideration these seasonal variations, the methane flux estimate may reach 8,850 mmol m$^{-2}$ y$^{-1}$. In the latter case, no-bloom, moderate-bloom and extreme-bloom conditions represent about 2 %, 5 %, and 93 % of the yearly $CH_4$ emissions, respectively. This estimate highlights the importance of $O_2$ microbubbles on the annual methane emission, a process not considered in conventional Fickian diffusion calculations (McGinnis et al., 2015), and suggests the need to better define during which periods of the year, under what weather conditions, with what bloom magnitude, the $O_2$ bubble point is exceeded. An estimate of the $CO_2$ consumption from green water lakes is about 1,140 mmol m$^{-2}$ y$^{-1}$, distributed in 28 %, 10 %, and 62 % during no-, moderate- and extreme-bloom conditions, respectively. Similarly, it is of about 1976 µmol m$^{-2}$ y$^{-1}$ for $N_2O$, distributed in 18%, 44% and 38%.

- It is not clear how could authors interpret from their results the influence on early rainfall in this subsection of discussion. The ms show same lakes before and after the rainy season. Therefore, this discussion might be possible, but the authors do not explore their results.

Actually, it is not exactly the same conditions. When the bloom disappears, the lake evolves back in a situation similar to the "no-bloom" conditions, except the pH in the sediments that is lower of about 2 units. It is possible that the GHG fluxes are impacted by the drop in pH. We therefore mention at the end of the manuscript, in the section "future directions" that measures will have to be carried out specifically after the disappearance of the bloom.

- In order to better address the upscaling, authors should clarify the source of the number of days without and with moderate or intense phytoplankton blooms (e.g. do they have any own dataset or only visual impression from these lakes? Or other source?).

OK, it comes from our own observations. We have now mentioned in the ms: "Based on 4 years of observation (2012-2015), a year can be divided into approximately 200 days without bloom throughout the rainy season, 100 days with moderate phytoplankton bloom during the dry season, and 65 days with a bloom magnitude sufficient for the $O_2$–supersaturation to be reached for 3 hours per day."

In addition, authors did not discuss any role of the observed daily variation on the upscaling.

The consequences of the observed daily variation are now in the discussion (see above "…and suggests the need to better define during which periods of the year, under what weather conditions, with what bloom magnitude, the $O_2$ bubble point is exceeded."

Since upscaling is not really the topic of this article, but will be developed in the future, this aspect is also addressed in the conclusion of the manuscript. "The difference in gas fluxes among the type of lake implies that it will be necessary to resort to remote sensing tools capable of discriminating them, but also to monitor the development of the phytoplankton bloom throughout the season for

any perspective of a regional GHG contribution estimate from surface water to the atmosphere. Lakes with green and black waters are the most common among Nhecolândia's alkaline lakes, but there are also red- and crystalline-water alkaline lakes, not to mention a wide range of freshwater lakes for which few data are available. In any case, all these lakes cannot be treated as one or two functional types."

MINOR COMMENTS
- Authors should revise the confusion related to the term "algae blooms", as other primary producer considered important (cyanobacteria) is not algae. A better term might be phytoplankton bloom. They should revise this term over the whole text.

It has been changed to "Phytoplankton bloom" throughout the ms.

- References are lacking in the analytical methods (e.g. page 5, section 2.2.2), and a fully revision is still needed for each method.

There was indeed a lack of reference to biomass measurements by fluorescence. However, this measure giving little information, since it was most of the time over-range, was removed from the manuscript as suggested by other referees.

- Authors should include a point after the term "wetland" in the page 9 (line 11).

OK, done.

- The sentence "Consistent with Martins (2012). (…) throughout the season" (page 9, lines 7-8) should be rephrased to anything like "Our results confirmed previous evidences on the different functioning of black and green water alkaline lakes (Martins, 2012), … "

It has been rephrased as suggested.

- What season do you mean in this complement "despite a very close mineral chemistry throughout the season" (page 9, line 8)?

Actually, it was throughout the seasons. Changed to "… despite a very similar mineral chemistry throughout the seasons."

- Page 10 (line 12): "strong sunshine" or "high incidence of solar radiation"?

OK, it has been changed

- The terms "fast or quick calculations" over the text should be replaced to anything like "rough estimates" (e.g. page 11, lines 5 or 14-15).

OK, Changed, see above reply.

- Authors should clarify that this ms assessed the variability over time and not spatially within the lake. They might complement the sentence "within the lake" to "within the lake over the daily cycle and year seasons".

OK it has been included.

- It's vague the sentence "…to estimate GHG emissions" (page 11, lines 20-21). What do authors mean? A regional or a global context? The GHG emissions of one of the freshwater wetland of the world? In addition, this conclusion without any argument seems speculative.

We mentioned "to estimate lakes annual GHG emissions"

- Figure 1: Images need scale and source.

Scale and sources have been incorporated. See below at the bottom of Fig. 1

[Figure]

**Figure 1: Location of the Pantanal wetland, Nhecolândia region, Nhumirím and Centenario farms and studied lakes. Satellite images are from Google EarthTM (bar = 1 km).**

- Figure 4:  What exactly means filled and dashes lines or the arrow?   This kind of description should be also in the legend.

The content in caption has been updated. See below:

[Figure]

**Figure 3: Oxidation – reduction potential and pH conditions in lake sediments. Note the drop in the pH value (arrow) occurring from September 10 to 11, 2015, in lake G after rainfall and disappearance of the cyanobacterial bloom.**

- Figures 6 and 7: Air-water fluxes and dissolved concentrations of a given gas should be in a same figure with two panels

We agree. Figures have been grouped into a single one with two panels.  See below:

[Figure]

**Figure 5: (a) Dissolved methane concentrations at the top of the water column, (b) and methane fluxes over 24 hours monitoring in black water lake (lake P) and green water lakes for no- (lake M), moderate- (lake V) and strong- (lake G) bloom conditions. Due to the logarithmic scale used, some negative values of the error bars (denoting standard deviations) are not drawn. The dashed line represents the beginning of the ebullition in lake G (13:20).**

- Figure 9: I did not understand why both CO2 and CH4 of the lake P is separated in another figure? Authors should organize all data in a same way among figures.

We agree. Black and green water lake results have been drawn on the same figure, see for example for CO2:

[Figure]

**Figure 7: Daily cycle of carbon dioxide fluxes showing emission from black water lake (P), and increasing consumption with increasing magnitude of the cyanobacterial bloom in green water lakes for no- (lake M), moderate- (lake V) and strong (lake G) bloom conditions.**